# Single-cell absolute contact probability detection reveals chromosomes are organized by multiple low-frequency yet specific interactions

Diego I. Cattoni [1], Andrés M. Cardozo Gizzi [1], Mariya Georgieva[1], Marco Di Stefano[2,3,4,5], Alessandro Valeri[1], Delphine Chamousset[1], Christophe Houbron[1], Stephanie Déjardin[1,6], Jean-Bernard Fiche[1], Inma González[6,8], Jia-Ming Chang[6,7], Thomas Sexton[6,9], Marc A. Marti-Renom [2,3,4,5], Frédéric Bantignies[6], Giacomo Cavalli [6] & Marcelo Nollmann[1]

At the kilo- to megabase pair scales, eukaryotic genomes are partitioned into self-interacting modules or topologically associated domains (TADs) that associate to form nuclear compartments. Here, we combine high-content super-resolution microscopies with state-of-the-art DNA-labeling methods to reveal the variability in the multiscale organization of the *Drosophila* genome. We find that association frequencies within TADs and between TAD borders are below ~10%, independently of TAD size, epigenetic state, or cell type. Critically, despite this large heterogeneity, we are able to visualize nanometer-sized epigenetic domains at the single-cell level. In addition, absolute contact frequencies within and between TADs are to a large extent defined by genomic distance, higher-order chromosome architecture, and epigenetic identity. We propose that TADs and compartments are organized by multiple, small-frequency, yet specific interactions that are regulated by epigenetics and transcriptional state.

[1] Centre de Biochimie Structurale, CNRS UMR5048, INSERM U1054, Université de Montpellier, 29 rue de Navacelles, 34090 Montpellier, France. [2] CNAG-CRG, Centre for Genomic Regulation (CRG), Barcelona Institute of Science and Technology (BIST), Baldiri i Reixac 4, 08028 Barcelona, Spain. [3] Gene Regulation, Stem Cells and Cancer Program, Centre for Genomic Regulation (CRG), Dr. Aiguader 88, 08003 Barcelona, Spain. [4] Universitat Pompeu Fabra (UPF), Barcelona 08010, Spain. [5] ICREA, Pg. Lluís Companys 23, 08010 Barcelona, Spain. [6] Institut de Génétique Humaine, CNRS UMR 9002, Université de Montpellier, 141 rue de la Cardonille, 34396 Montpellier, France. [7] Department of Computer Science, National Chengchi University, 11605 Taipei City, Taiwan. [8] Epigenetics of Stem Cells, Department of Stem Cell and Developmental Biology, Institut Pasteur, CNRS UMR3738, 25 rue du Docteur Roux, 75015 Paris, France. [9] Institut de génétique et de biologie moléculaire et cellulaire, CNRS UMR 7104 - Inserm U 964, 67404 Illkirch, France. Diego I. Cattoni, Andrés M. Cardozo Gizzi and Mariya Georgieva contributed equally to this work. Correspondence and requests for materials should be addressed to M.N. (email: marcelo.nollmann@cbs.cnrs.fr)

The multiscale organization of eukaryotic genomes defines and regulates cellular identity and tissue-specific functions[1–3]. At the kilo-megabase scales, genomes are partitioned into self-interacting modules or topologically associated domains (TADs)[4–6]. TAD formation seems to require specific looping interactions between TAD borders[7, 8], while the association of TADs can lead to the formation of active/repressed compartments[9]. These structural levels were often seen as highly stable over time; however, recent single-cell Hi-C studies have reported different degrees of heterogeneity[10, 11]. Other studies have reported that genomes also display stochasticity in their association with the nuclear lamina[12], in the formation of chromosome territory neighborhoods[13], and in gene kissing[14]. However, access to single-cell absolute probability contact measurements between loci and efficient detection of low-

frequency, long-range interactions are essential to quantify the stochastic behavior of chromatin at different scales.

Here, we combined high-content super-resolution microscopy with state-of-the-art DNA-labeling methods to reveal the variability in the multiscale organization of chromosomes in different cell types and developmental stages in *Drosophila*. Remarkably, we found that stochasticity is present at all levels of chromosome architecture, but is locally modulated by sequence and epigenetic state. Contacts between consecutive TAD borders were infrequent, independently of TAD size, epigenetic state, or cell type. Moreover, long-range contact probabilities between non-consecutive borders, the overall folding of chromosomes, and the clustering of epigenetic domains into active/repressed compartments displayed different degrees of stochasticity that globally depended on cell type. Overall, our results show that contacts

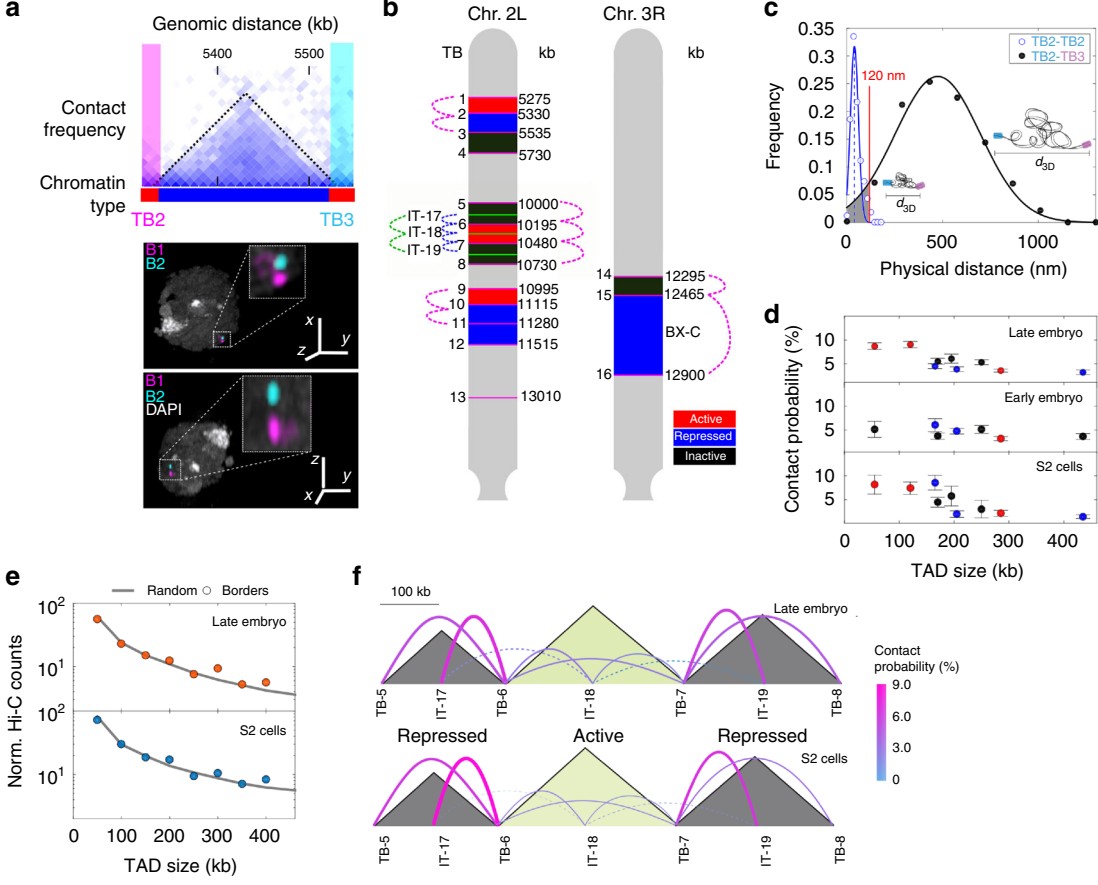

**Fig. 1** TAD organization arises from modulation of stochasticity. **a** Top, region of Hi-C contact matrix of chromosome 2L. The black-dotted line demarcates a TAD and pink and cyan boxes represent the Oligopaint- labeled TAD borders (TB). Chromatin epigenetic state is indicated at the bottom using the color code of panel **b**. Bottom, representative three-color 3D-SIM image in two orientations. DAPI, TB2, and TB3 are shown in gray, pink, and cyan, respectively. Scale bar = 1 μm for the main image. The inset displays 5× amplification of the selected region. **b** Oligopaint libraries in chromosomes 2L and 3R employed in this study (TB1-16 at TAD borders and IT17-19 within TADs). Colored boxes display the chromatin type of TADs as defined in Supplementary Fig. 1a, b. Red: active, blue: repressed, and black: inactive. Dotted colored lines indicate the combinations of libraries measured. **c** 3D distance distributions between TB2–TB2 and TB2–TB3. The mean colocalization resolution, estimated from two-color labeling of a single border (40 nm, vertical blue dashed line). Blue and black solid lines represent Gaussian fittings. The absolute contact probability between libraries was obtained from the integral of the area of the Gaussian fitting (shaded gray) below 120 nm (Supplementary Fig. 1e). $N = 161$ and 556 for TB2–TB2 and TB2–TB3, respectively, from more than three biological replicates. **d** Absolute contact probability between consecutive borders vs. genomic distance. Chromatin state of TADs is color coded as defined in panel 1b. Error bars represent SEM. **e** Normalized Hi-C counts between consecutive TAD borders (circles) and random loci (solid gray line) as a function of genomic distance for S2 and late embryonic cells. Matrix resolution = 10 kb. Two biological replicates for each cell type were performed. **f** Schematic representation of contact probability between and within TADs (solid colored lines) for late embryo and S2 cells at the chromosomal region shaded in panel **b**. Sizes of TADs (gray-shaded triangles) are proportional to genomic length (scale bar on top). Chromatin type is indicated at the bottom. The thickness of the lines and color indicate the absolute contact probability. Dotted lines indicate inter-TAD contacts. Early embryo measurements are depicted in Supplementary Fig. 1k. Numbers of cells for each combination are provided in Supplementary Fig. 1f–h

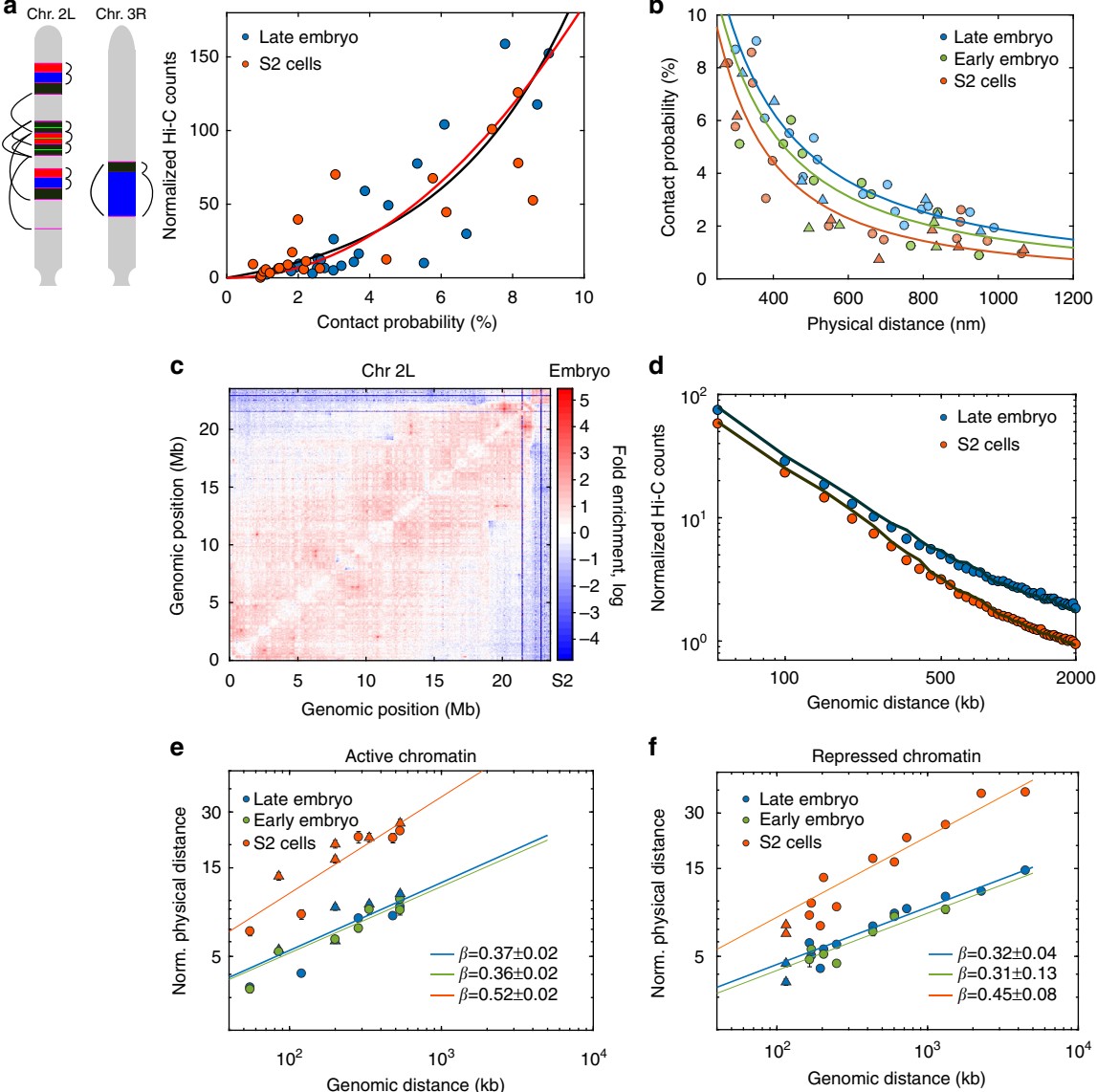

**Fig. 2** Long-range absolute contact probability is specifically modulated for each cell type. **a** Left, a schematic representation of pairwise distance measurements between consecutive and nonconsecutive borders, with color code and positions as in Fig. 1b. Right, normalized Hi-C counts vs. microscopy absolute contact probability for consecutive and nonconsecutive domain borders for embryo and S2 cells. Solid black and red lines represent exponential and power-law fits, respectively. Matrix resolution = 10 kb. N for microscopy pairwise measurements is provided in Supplementary Fig. 1f–h. N = 2 for Hi-C data, from at least three and two biological replicates, respectively. **b** Absolute contact probability vs. mean physical distance between probes for consecutive and nonconsecutive TAD borders (filled circles). Solid lines represent power-law fittings with the scaling exponent described in Supplementary Fig. 2b. Triangles represent measurements within TADs. **c** Matrix of relative frequency of normalized Hi-C counts for late embryo vs. S2 cells for chromosome 2L. Contact frequency ratio is color coded according to scale bar. Matrix resolution = 50 kb. N = 4, biological replicates. **d** Log–log plot of normalized Hi-C counts between TAD borders vs. genomic distance for embryo and S2 cells. Solid lines represent the average contact frequency for randomly chosen positions in the genome. Matrix resolution = 10 kb. N = 2, biological replicates. **e,f** Log–log plot of the mean physical distance vs. genomic length for (**e**) active and (**f**) inactive/repressed chromatin domains for different cell types. Mean distance values were normalized by the pre-exponential factor from the power-law fit of each data set (Supplementary Fig. 2d, e). Solid lines show the power-law fits, with the scaling exponent β shown in the panel. Circles and triangles are depicted as described in panel 2b. Error bars represent SEM. N > 140 for each data point, from more than three biological replicates (Supplementary Fig. 1)

between and within TADs are rare, but can be epigenetically modulated to give rise to different levels of higher-order genome organization. We anticipate that our results will guide new statistical models of genome architecture and will be a starting point for more sophisticated studies to understand how a highly variable, multiscale organization can ensure the maintenance of stable transcriptional programs through cell division and during development.

## Results

**Multiple low-frequency interactions mediate TAD assembly.** A major mechanism for TAD formation in mammals involves the stable looping of TAD borders[8]. Stable looping between TAD borders was also recently proposed to be relevant for the maintenance of transcriptional programs during *Drosophila* development[7]. However, long-lived stable interactions are unlikely to allow for rapid responses in gene regulation. To study this

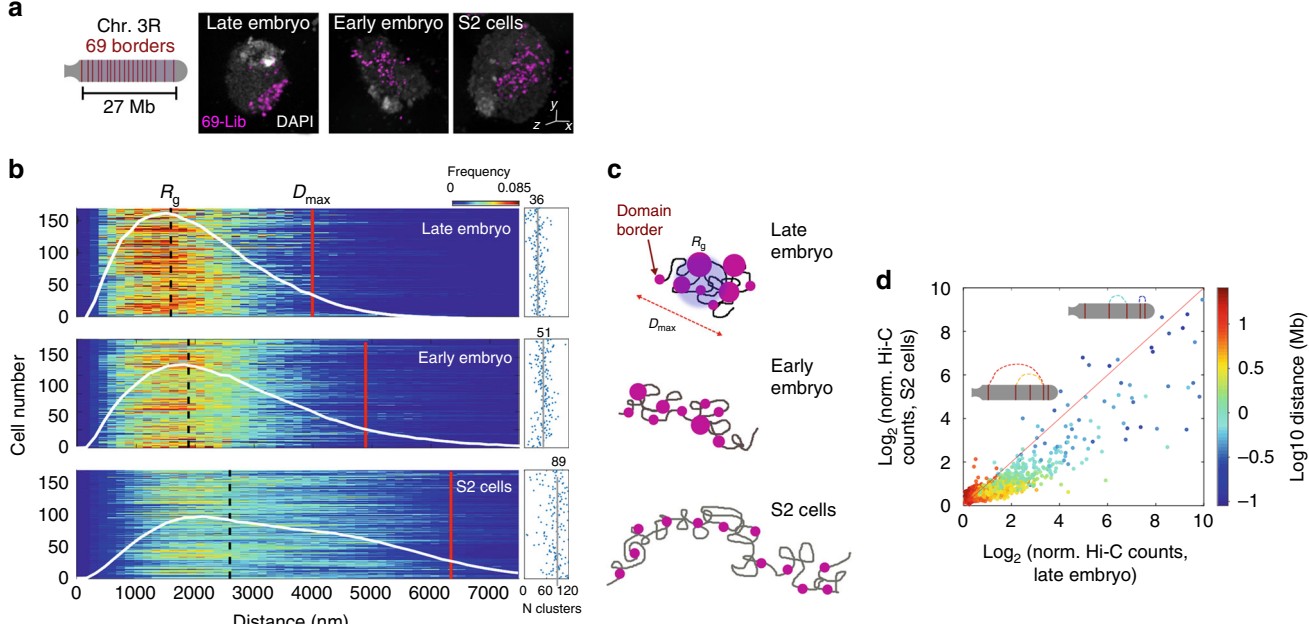

**Fig. 3** Cell-type-specific frequency of long-range contacts defines chromosome folding in 3D space. **a** Left, schematic representation of 69 domain borders labeled by a single Oligopaint library (Lib-69) in Chr. 3R. Each probe was spanned at ~20 kb, and probes were separated by 320 kb on average (Supplementary Fig. 3a, b). Right, representative two-color 3D-SIM images for all studied cell types. DAPI signal (white) and Lib-69 (pink) are shown. Scale bar = 200 nm. **b** Left panel, single-cell probability distance distribution $p(r)$ between all pairs of foci imaged by 3D-SIM. The white line represents the population-averaged $p(r)$ frequency. Detailed $R_g$ and $D_{max}$ values are shown in Supplementary Fig. 3. $D_{max}$ is defined as the distance that comprises <97% of the area under the $p(r)$ function. Right panel, number of foci per cell for each condition with mean population values shown as solid vertical lines and indicated above. $N = 180$, from more than three biological replicates. **c** Schematic representation of the chromosome structure for each cell type. The solid gray line represents the chromatin fiber and pink circles represent domain borders with sizes proportional to the number of regrouped borders. **d** Hi-C contact frequencies of S2 vs. late embryo cells for all the pairwise combinations of the 69 borders. The solid red line represents the relation expected if frequencies of interactions between the 69 borders were equal between cell types. Insets depict chromosome 3R and different combinations of genomic distances and frequencies of interaction between borders. Matrix resolution = 50 kb. $N = 4$, from at least three biological replicates

apparent contradiction, we developed a method to dissect the changes in TADs organization at the single-cell level in three transcriptionally distinct *Drosophila* cell types: early (stage 5) and late (stage 16) embryos; and an immortalized cell line (S2). Pairs of TAD borders were labeled with Oligopaints libraries[15] and imaged using multicolor three-dimensional structured illumination microscopy (3D-SIM)[16, 17] (Fig. 1a). TAD chromatin types were defined as active, repressed, or inactive following the distribution of epigenetic marks (Supplementary Fig. 1a). Borders flanking TADs with different chromatin states were imaged in chromosomes 2L and 3R (Fig. 1b and Supplementary Fig. 1b), and appeared in microscopy as well-defined foci (Fig. 1a) whose size increased proportionally with the genomic length of the library (Supplementary Fig. 1c). A large proportion of cells (60–70%) displayed a single foci, consistent with a high degree of homologous pairing independently of the ploidy of each cell type (Supplementary Fig. 1d)[18, 19]. Distances between TAD borders were Gaussian distributed for all cell types (Fig. 1c and Supplementary Fig. 1f–h). Remarkably, the width of these distributions was comparable to the mean distance between TAD borders, revealing a high degree of structural variability, independently of TAD size or epigenetic state (Fig. 1c and Supplementary Fig. 1i). Further, the linear relation between dispersion and physical distance (Supplementary Fig. 1i-j) suggests that this variability is regulated by the polymer properties of the chromatin fiber.

Next, we quantified the absolute contact probability between consecutive borders by integrating the probability distance distributions below 120 nm (99% confidence interval obtained from single-library two-color control experiments, Fig. 1c and Supplementary Fig. 1e). Notably, the contact probability between

consecutive TAD borders was below 10%, independently of the cell type or of the epigenetic state of the TAD being flanked (Fig. 1d). Consistently, Hi-C contact frequencies between consecutive TAD borders vs. random genomic loci were indistinguishable (Fig. 1e). These results, combined with the lack of enrichment of CTCF and cohesin at TAD borders in *Drosophila*[20], suggest that TAD assembly does not involve stable loops in flies, but rather can be explained by an "insulation–attraction" mechanism[21]. This model may provide an alternative explanation for the formation and maintenance of more than 50% of metazoan TADs whose boundaries are not formed by looping interactions as defined by Hi-C experiments[8].

In agreement with this model, absolute contact probabilities within TADs and between their borders were similar (Fig. 1f and Supplementary Fig. 1k), with inactive/repressed TADs displaying higher contact probabilities than active TADs (7 ± 1% vs. 2.7 ± 1%, mean ± SD). Contact probabilities within TADs were in all cases considerably higher than those with neighboring TADs (Fig. 1f), indicating that stochasticity is locally modulated at the TAD level. Of note, contacts across TAD borders were not uncommon (~3%, Fig. 1f), implying frequent violations of boundary insulation at TAD borders. These results indicate that confinement of chromatin into TADs may require only small differences in absolute contact probabilities (~2-fold). Thus, condensation of chromatin into TADs may arise from a multitude of low-frequency, yet specific, intra-TAD contacts.

**Infrequent long-range contacts modulate chromatin folding.** Recent Hi-C studies suggested that stable clustering between neighboring active TAD borders regulates transcriptional

programs that persist during development[7]. We directly tested this hypothesis by measuring the contact probabilities between nonconsecutive TAD borders (Fig. 2a). Hi-C contact frequencies among TAD borders increased nonlinearly with absolute contact probabilities (Fig. 2a and Supplementary Fig. 2a), with both exponential and power-law empirical models fitting the data equally well. Our results highlight the ability of Hi-C to enhance the detection of high-probability contacts and also suggest the need to relate Hi-C data to physical distances with a nonlinear relationship. This would allow a better discrimination of

low-frequency contacts (1–3%, Fig. 2a) such as those observed within and between TADs (Fig. 1f) and a more realistic conversion of Hi-C maps into 3D-folded structures.

Contact probabilities between nonconsecutive TAD borders were in all cases low (<9%, Fig. 2b) and decreased monotonically with physical and genomic distance following a power-law behavior (Fig. 2b and Supplementary Fig. 2b–c). Notably, the decay exponents were different between cell types (Fig. 2b and Supplementary Fig. 2b), indicating that levels of stochasticity are globally modulated between cell types, possibly reflecting cell

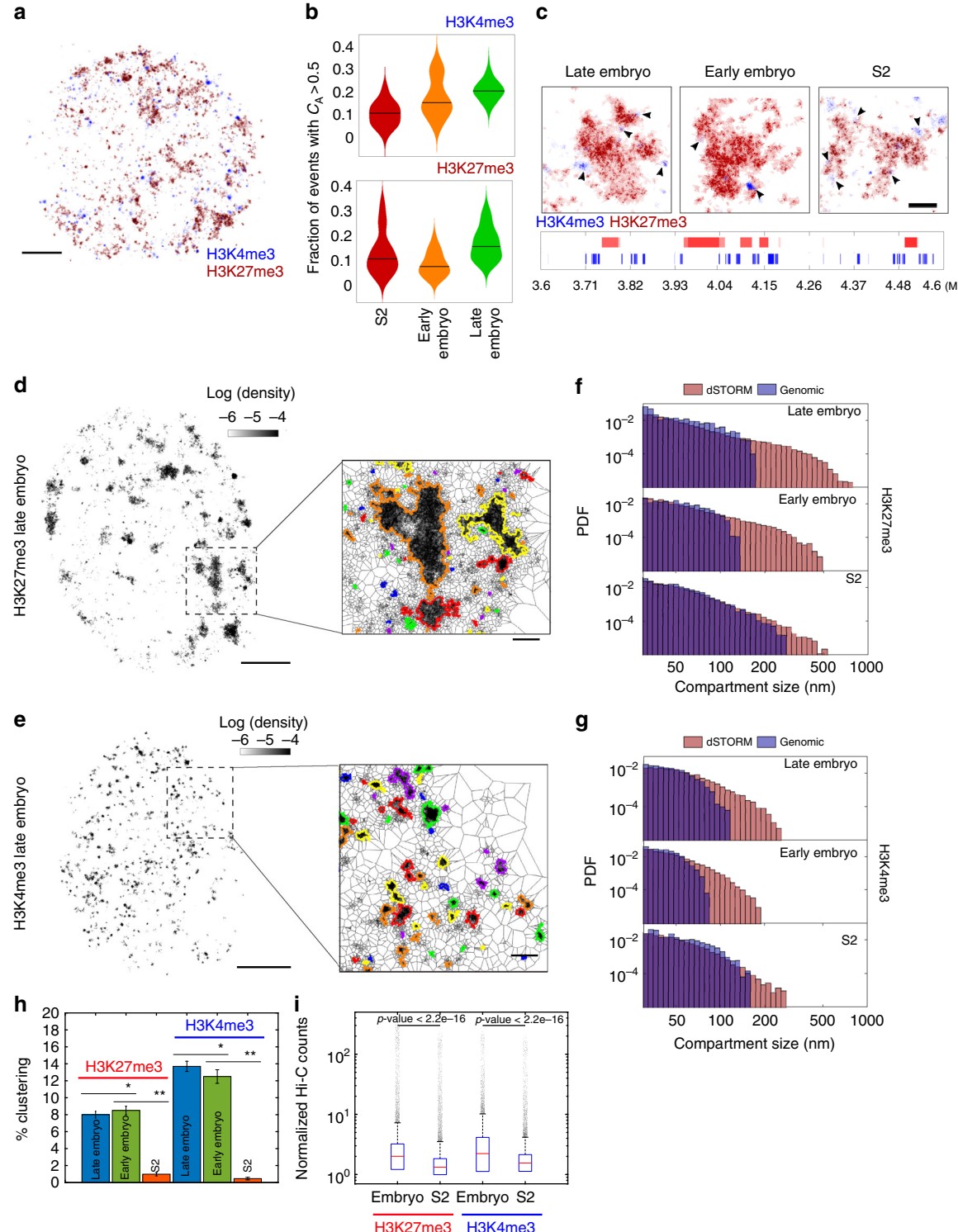

type-specific transcriptional programs. To test whether this tendency was held genome-wide, we calculated the ratio between normalized Hi-C contact maps of embryos and S2 cells. For all chromosomes, embryos displayed a higher relative contact frequency than S2 cells below a few Mb (Fig. 2c and Supplementary Fig. 2d), in accordance with our microscopy results. Furthermore, the frequency of contacts between non-consecutive TAD borders genome-wide was similar to that of random genomic loci for both cell types (Fig. 2d). As the large majority of TAD borders in *Drosophila* contain active chromatin[22, 23], our results are inconsistent with stable preferential looping of active borders[7, 24], and rather indicate that these contacts are rare or short-lived. This interpretation is consistent with the transient assembly and disassembly of transcription clusters in human cells[25].

Next, we sought to determine if this modulation in contact probabilities resulted from cell-type- specific changes in the local folding properties of the chromatin fiber. To this end, we measured the end-to-end distance ($d_{3D}$) for active or inactive/repressed chromatin domains of varying genomic ($d_{kb}$) lengths. For all cell and chromatin types, we observed a power-law scaling behavior ($d_{3D} \alpha d_{kb}$) (Fig. 2e, f and Supplementary Fig. 2e–f) with scaling exponents being higher for active than for repressed domains, consistent with previous measurements in Kc$_{167}$ cells[26]. Theoretical studies of polymer physics suggest that the exponent of polymers with random coil behavior is 1/2, while that of an equilibrium globule is 1/3 (Mirny 2011). Thus, our power-law exponents situate between these two extremes, suggesting an intermediate behavior. Remarkably, scaling exponents were considerably lower in embryos than in S2 cells, for both types of chromatin. TAD border localization is conserved between cell types[22, 27]; however, our results show that TAD conformation and structural heterogeneity strongly depend on cell type. This cell-type specificity in TAD organization may result from the interplay between the degree of chromatin compaction and the frequency of stochastic long-range interactions.

**Impact of long-range contacts in chromosome-wide folding**. To quantitatively dissect stochasticity at larger genomic scales, we labeled 69 quasi-equidistant TAD borders encompassing 90% of chromosome 3R (Fig. 3a and Supplementary Fig. 3a, b). Tens of foci were resolved in embryonic and S2 cells by 3D-SIM (Fig. 3a). The paired probability distance distribution $p(r)$ between any two foci exhibited moderate single-cell variations (Fig. 3b) but was considerably different between cell types (Fig. 3b and Supplementary Fig. 3c). The chromosome elongation and mean volume, obtained from the maximum pairwise distance ($D_{max}$) and the

radius of gyration ($R_g$, Fig. 3c), decreased to almost half when comparing S2 and late embryonic cells, while early embryonic cells adopted intermediate values (Fig. 3b).

From the number of labeled barriers (69) and the pairing frequency of homologous chromosomes (Supplementary Fig. 1c), we can estimate a maximum of 90–100 resolvable foci/cell in the absence of any long-range interactions (Supplementary Fig. 3). Our imaging results show an average of 89 ± 28 foci/cell for S2 cells (mean ± SD, Fig. 3b), confirming our predictions and consistent with a very low frequency of long-range interactions for this cell type (see discussion in Supplementary Fig. 3). Surprisingly, in early and late embryos, the number of observed foci was considerably reduced (51 ± 20 and 36 ± 13, respectively, mean ± SD, Fig. 3b), revealing higher probabilities of long-range interactions for these cell types. The lower number of foci detected was not associated with the smaller volume of embryonic cell nuclei, causing the probes to be closer than the resolution limit of 3D-SIM microscopy (Supplementary Fig. 3e). Furthermore, for each cell type, the number of foci displayed a very low or nonexistent correlation with nucleus size (Supplementary Fig. 3f).

Reinforcing these findings, changes in Hi-C contact frequency of S2 vs. late embryo for the 69 TAD borders were notable in the sub-Mb scale (200–600 kb), and they extended to genomic distances as high as ~10 Mb (Fig. 3d), suggesting that changes in chromosome compaction between cell types arise from an increased frequency of interactions affecting all genomic scales. All in all, these data indicate that chromosome folding is highly variable, with mild, cell type-specific increases in the probability of long-range contacts being sufficient to produce large changes in the manner in which chromosomes occupy the nuclear space (Fig. 3c).

**Stochastic nanoscale organization of epigenetic marks**. Inter-chromosomal and intrachromosomal Hi-C maps have revealed that active and repressed TADs may associate to form two types of compartments (namely A and B)[9, 28]. To study this higher-order level of organization in single cells and at the single-molecule level, we immunolabeled active and repressive epigenetic marks (histones H3K4me3 and H3K27me3, respectively) and performed multicolor direct stochastic optical reconstruction microscopy (dSTORM)[29–31], a method that provides a higher spatial resolution than 3D-SIM. dSTORM imaging revealed that active and repressive histone marks distributed non-homogeneously across the cell nucleus, forming discrete compartments of tens to hundreds of nanometers for all cell types (Fig. 4a and Supplementary Fig. 4a). Repressed and active

**Fig. 4** Chromatin reorganization between cell types is modulated by stochastic clustering between epigenetic domains. **a** Two-color dSTORM image of active (H3K4me3, blue) and repressive (H3K27me3, red) chromatin marks in a representative S2 cell. Images of early and late embryos are displayed in Supplementary Fig. 4a and panel c. Scale bar = 1 µm. **b** Quantification of co-occurrence (CA > 0.5) between active and repressive chromatin using aCBC[32]. Violin plots of CAs for H3K4me3 and H3K27me3 are shown in the upper panel and lower panels, respectively. The black line represents the median of the distribution. **c** Representative zoomed images of two-color dSTORM for the three cell types investigated. Black arrows indicate the localization of small active chromatin domains in the periphery of large repressive domains. The lower panel displays active and repressive marks of Chip-Seq enrichment profiles for late embryo. Scale bar = 200 nm. **d,e** dSTORM-rendered images of Alexa-647-labeled **d** H3K27me3 and **e** H3K4me3. Images show density maps computed from the area of the polygons obtained from the Voronoï diagram with scale defined on top. Scale bar = 1 µm. Zoomed regions display detected compartments (highlighted with different colors). Scale bar = 200 nm. Additional images for all cell types and chromatin marks are displayed in Supplementary Fig. 5a, b. **f,g** Population-based distribution of epigenetic domain sizes as obtained from dSTORM and predicted from ChIP-seq data for H3K27me3 **f** and H3K4me3 **g**. PDF is probability density function. Single-cell distributions of physical sizes and Chip-Seq data are shown in Supplementary Figs. 5c, d and 6b, respectively. N = 60, from two to three biological replicates in microscopy imaging. **h** Percentage of clustering for active and inactive chromatin marks for each cell type. Error bars = SD. One-sample *t* test *p*-values: *$p < 0.01$; **$p < 0.001$. **i** Box plots of the distributions of normalized Hi-C counts between chromatin domains of H3K27me3 or H3K4me3 in embryos and S2 cells. The results were independent of matrix resolution (10, 20, and 50 kb). Boxes contain 50% of the data ($0.67\sigma$), and red lines mark the median values. Outliers (>$3.3\sigma$ away from the mean values) are shown as black dots. *p*-values were calculated using the Welch *t* test

chromatin marks were strictly segregated at the nanoscale for all cell types, as revealed by coordinate-based colocalization analysis (aCBC[32], Fig. 4b). These findings were confirmed by independent colocalization methods and by additional controls using doubly labeled nuclear factor and noncolocalizing epigenetic marks (Supplementary Fig. 4b–d). Interestingly, active marks were often observed at borders of/or demarcating large repressed compartments, mirroring their alternating one-dimensional genomic distributions (Fig. 4c).

To investigate if active and repressed compartments also varied among cell types and development, we resorted to one-color dSTORM using Alexa 647 as the fluorophore of choice (the results were similar when using other fluorophores, Supplementary Fig. 4e). Compartments were detected using a Voronoi diagram-based algorithm (Fig. 4d, e)[33]. In all cases, active compartments were smaller than repressive compartments in agreement with two-color dSTORM observations (Fig. 4c–e and Supplementary Fig. 5a, b). Interestingly, for both marks, the number of compartments and their sizes showed variations between single cells of the same type (Supplementary Fig. 5c, d). To further evaluate if changes in compartment sizes correlated with changes in local chromatin folding, we quantified the density of single-molecule detections in active and repressed compartments. Notably, the local density of compartments was higher for both types of marks in embryonic cells than for S2 cells (Supplementary Fig. 6a), consistent with our previous findings (Fig. 2e, f) and with compartment contact density from Hi-C counts (Supplementary Fig. 6b).

To study whether the nanoscale organization of repressive and active marks reflected the epigenomic domain organization from ensemble genome-wide methods, we predicted the physical sizes of epigenomic domains (Supplementary Fig. 6c) and compared them with those obtained by direct observation. The predicted size distributions failed to recover the largest compartments observed by microscopy (Fig. 4f, g and Supplementary Fig. 6c). We reasoned that large compartments are likely to arise from clustering of smaller epigenetic domains ("clustered compartments").

To quantify this phenomenon, we calculated the percentage of compartments not accounted for by the distribution of epigenetic domains. This percentage of clustered compartments was below <10% for embryonic cells and almost absent in S2 cells (Fig. 4h). The latter is consistent with higher Hi-C contact frequency between H3K27me3 domains in embryos than in S2 cells (Fig. 4i). Repressive and active compartments showed different degrees of clustering (Fig. 4f–h), indicating that stochasticity can be specifically modulated by transcriptional/epigenetic states. This is likely due to the different mechanisms of clustering formation at play, such as Polycomb regrouping of repressed genes[34] vs. transient interactions of active genes[35, 36]. It is important to note, however, that the large majority of compartments (~90%) could be accounted for by the predicted distributions of epigenomic domains, consistent with the majority of the epigenetic domains described by genome-wide methods existing at the single-cell level. These results are consistent with the cell type-specific higher-order organization of chromatin arising from stochastic contacts between chromosomal regions harboring similar epigenetic marks, likely reflecting cell type-specific transcriptional programs.

## Discussion

In this work, we show that genome organization in *Drosophila* is not driven by stable or long-lived interactions but rather relies on the formation of transient, low-frequency contacts whose frequencies are modulated at different levels. Stochasticity is modulated locally at the TAD level by specific intra-TAD interactions, and globally at the nuclear level by interactions of TADs

of the same epigenetic type. Furthermore, stochasticity is also regulated between cell types. These modulated stochasticities reveal a novel mechanism for the spatial organization of genomes. These pieces of evidence could be critical for a more accurate understanding of how different cell types interpret genomic and epigenomic states to produce different phenotypes. Dynamic measurements of chromosome organization with high coverage will be needed in future to further explore the origin of heterogeneity in chromosome architecture and to determine whether genome organization is a stationary or a fully stochastic process.

In mammals, a large proportion of consecutive TAD borders is looped by specific interactions apparently mediated by CTCF and cohesin[8, 37, 38]. Recent reports suggested that this mechanism may also be at play in *Drosophila*[7, 24]. Our results, however, provide compelling evidence that looping of consecutive TADs borders in *Drosophila* is rare at the single-cell level. These observations, supported by recent studies showing that cohesin-enriched loop anchors in *Drosophila* are found within TADs rather than at TAD borders[39, 40], are against TAD boundaries being the bases of stable chromatin loops. Thus, the lack of frequent interactions between TAD borders could be consistent with a model where TADs arise from a dynamic balance between cohesin-mediated loop extrusion[41], the blocking of that movement by architectural proteins, and factors that may load or remove cohesin[42, 43]. In *Drosophila*, however, CTCF and cohesin are not found enriched at TAD borders. Thus, we envision that other factors (e.g., Beaf-32 and CP190/chromator instead of CTCF and cohesin) could play a role at looping and dynamically extruding distant DNA fragments within the same TAD. In addition, active marks may help determine the properties of TAD boundaries[23,] while other epigenetic marks could play a role in the formation of polycomb and inactive TADs[44]. Similar epigenetic mechanisms may even play a role in TAD folding in mammals, consistent with the observation that CTCF depletion leads only to minor changes in TAD organization[45]. Importantly, our data provide quantitative estimates of the stochasticity and absolute frequencies of interactions within TADs, imposing important constraints on any model of TAD formation in *Drosophila*.

Recent reports suggested that TAD borders enriched in housekeeping genes form stable 3D colocalization patterns that persist during development[7]. In contrast, we found that 3D contacts between TAD borders are rare and highly stochastic in all cell types investigated. These results are consistent with recent single-nucleus-Hi-C studies reporting that TAD formation is highly stochastic in mammals[11], and with the rapid association and dissociation of transcription foci[25] rather than with stable transcription factories.

Most current spatial models of genome architecture rely on interpreting interaction maps from chromosome conformation capture-based experiments, which seize the relative frequencies of interactions between loci at close spatial proximity. However, translation of relative contact frequencies into spatial distances is challenging. Our direct single-cell measurements of absolute contact probabilities, full distance distributions, and dissection of low-frequency events for different chromatin and cell types will complement the existing methods to refine the next generation of statistical models of genome architecture. Our results call for more sophisticated studies to reveal how a highly stochastic genome organization can ensure the maintenance of stable transcriptional programs through cell division and during development.

## Methods

**Cell culture and embryonic tissue preparation**. *Drosophila* S2 cells were obtained from the *Drosophila* Genomics Resource Center. S2 cells were grown in serum-supplemented (10%) Schneider's S2 medium at 25 °C. Oregon-R w[1118] fly stocks

were maintained at room temperature (RT) with natural light/dark cycle and raised in standard cornmeal yeast medium. Following a precollection period of at least 1 h, fly embryos were collected on yeasted 0.4% acetic acid agar plates and incubated at 25 °C until they reached the desired developmental stage: 2–3 h or 12–14 h (total developmental time) for early and for late embryos, respectively. Embryos were mechanically broken and immediately fixed by using 4% PFA in PBS for 10 min at RT[46]. S2 cells were allowed to adhere to a poly-L-lysine coverslip for 30 min in a covered 35-mm cell culture dish before 4% PFA fixation.

**Immunostaining.** Cells were permeabilized with 0.5% Triton X-100 for 10 min and blocked with 5% of bovine serum albumin (BSA) for 15 min at RT. Primary antibodies anti-H3K27me3 (pAb-195-050, Diagenode and ab6002, Abcam), anti-H3K4me3 (cat#04-745, Millipore and ab1012, Abcam), anti-Polycomb[47,] and anti-Beaf-32[48] (made from a rabbit by Eurogentec) were coupled to Alexa Fluor 647 or Cy3b, as described elsewhere[32]. Antibodies were used at a final concentration of 10 µg ml$^{-1}$ in PBS and 1% BSA. Coverslips were incubated overnight at 4 °C in a humidified chamber and washed three times with PBS before introducing fiducial markers diluted at 1/4000 (Tetraspeck, #10195142, FisherScientific). Coverslips were mounted on slides with 100-µl wells (#2410, Glaswarenfabrik Karl Hecht GmbH & Co KG) in dSTORM buffer composed of PBS, glucose oxidase (G7141-50KU, Sigma) at 2.5 mg ml$^{-1}$, catalase at 0.2 mg ml$^{-1}$ (#C3155-50MG, Sigma), 10% glucose, and 50 mM of β-mercaptoethylamine (MEA, #M9768-5G, Sigma). Coverslips were sealed with duplicating silicone (Twinsil, Rotec).

**Oligopaint libraries.** Oligopaint libraries were constructed from the Oligopaint public database (http://genetics.med.harvard.edu/oligopaints). All libraries consisted of 42-mer sequences discovered by OligoArray2.1 run with the following settings: -n 30 -l 42 -L 42 -D 1000 -t 80 -T99 -s 70 -x 70 -p 35 -P 80 -m "GGGG;CCCC;TTTTT; AAAAA" -g 44. Oligonucleotides for libraries 1–18 and BX-C were ordered from CustomArray (Bothell, WA). The procedure used to synthesize Oligopaint probes is described below. Chr3R-69 borders oligonucleotides were purchased from MYcroarray (Ann Arbour, MI). Oligopaint probes for this library were synthesized using the same procedure as for the other libraries except for the initial emulsion PCR step. Secondary, fluorescently labeled oligonucleotides were synthesized by Integrated DNA Technologies (IDT; Coralville, IA for Alexa488) and by Eurogentec (Angers, France for Cy3b). See Supplementary Table 1 for a list of Oligopaint probe sets used for libraries 1–18. Sequences for secondary oligonucleotides and PCR primers are described below (Supplementary Tables 2–4). Details for the methods used for probe synthesis are provided in Supplementary Notes.

**Fluorescence in situ hybridization.** To prepare sample slides containing fixed S2 cells for FISH, S2 cells were allowed to adhere to a poly-L-lysine coverslip for 1 h in a covered 35-mm cell culture dish at 25 C. The slides were then washed in PBS, fixed in 4% paraformaldehyde (PFA) for 10 min, rinsed 3 times for 5 min in PBS, permeabilized for 10 min with 0.5% Triton, rinsed in PBS, incubated with 0.1 M HCl for 10 min, washed 3 times for 1 min with 2× saline-sodium citrate—0.1% Tween-20 (2×SSCT), and incubated in 2×SSCT/50% formamide (v/v) for at least 30 min. Then, probes were prepared by mixing 20 µl of hybridization buffer FHB (50% formamide, 10% Dextransulfat, 2×SSC, and Salmon Sperm DNA 0.5 mg ml$^{-1}$), 0.8 µl of RNAse A, 30 pmol of primary probe, and 30 pmol of secondary oligo. An aliquot of 12 µl of this mix was added to a slide before adding and sealing with rubber cement the coverslips with cells onto the slide. Probes and cells are finally codenaturated for 3 min at 78 °C before hybridization overnight at 37 °C. The next day, the slides were washed three times for 5 min in 2× SSC at 37 °C, and then three times for 5 min in 0.1× SSC at 45 °C. Finally, they were stained with 0.5 µg ml$^{-1}$ of DAPI for 10 min, washed with PBS, mounted in Vectashield, and sealed with nail polish. For a more detailed protocol, see[49].

**Image acquisition and postprocessing of 3D-SIM data.** Samples were prepared as described above and mounted on an OMX V3 microscope (Applied Precision Inc.) equipped with a 100×/1.4 oil PlanSApo objective (Olympus) and three emCCD cameras. Laser lines at 405-nm, 488-nm, and 561-nm excitation were used to excite DAPI, Alexa488, and Cy3b, respectively. Each channel was acquired sequentially. A transmission image was also acquired to control for cell morphology. For each channel, a total of 1455 images made of 97 different Z-planes separated by 125 nm were acquired, in order to acquire a stack of 12 µm. Three different angles (60°, 0°, and + 60°). as well as five-phase steps were used to reconstruct 3D-SIM images using softWoRx v5.0 (Applied Precision Inc.). The final voxel size was 39.5 nm in the lateral (xy) and 125 nm in the axial (z) directions, respectively, for a final 3D stack volume of ~40 × 40 × 12 µm. Multicolor TetraSpeck beads (100 nm in diameter, Invitrogen) were used to measure x, y, and z offsets, rotation about the z-axis, and magnification differences between fluorescence channels. These corrections were applied to the reconstructed images. The same beads were used to validate the reconstruction process, ensuring a final resolution of ~120 nm in xy and ~300 nm in z at 525 nm of emission wavelength. 3D-SIM raw and reconstructed images were analyzed with SIMCheck ImageJ Plug-in[50]. Acquisition parameters were optimized to obtain the best signal-to-noise ratio, avoiding photobleaching between the different angular, phase, and axial acquisitions.

**3D nuclei segmentation from 3D-SIM data.** 3D-SIM images were analyzed employing homemade software written in Matlab. In order to identify nuclear shells, nuclei are first segmented by manually selecting rectangular regions of interest (ROIs) of the DAPI signal in the XY-plane and keeping all the Z-planes, and then, a low-pass filter is applied to the DAPI intensities, so that only the large-scale information (i.e., nuclear shape) is kept. For each plane of the 3D ROIs, an intensity threshold is computed, as described by Snell et al.[51] in order to distinguish voxels inside or outside the nucleus. The average intensity threshold calculated from the threshold of the single planes is used to identify the complete nuclear shell. After nuclei segmentation, foci were identified by calculating, for each channel separately, the maximum entropy threshold of the fluorescence intensities in the 3D ROIs. By using the intensity thresholds, the 3D ROIs are finally binarized (voxels above the threshold are set to 1, while the others are set to 0) and the different foci are identified as groups of connected voxels. From the group of connected voxels, the center of mass was estimated with subpixel resolution. The distance between TBs was estimated as the linear distance between the closest foci imaged in two different emission channels.

**Image acquisition of two-color dSTORM data.** Super-resolution experiments were carried out in a custom-made inverted microscope employing an oil-immersion objective (Plan-Apocromat, 100×, 1.4NA oil DIC, Zeiss) mounted on a z-axis piezoelectric stage (P-721.CDQ, PICOF, PI). For 2D imaging, a 1.5× telescope was used to obtain a final imaging magnification of 150-fold corresponding to a pixel size of 105 nm. Three lasers were used for excitation/photoactivation: 405 nm (OBIS, LX 405-50, Coherent Inc.), 488 nm (OBIS, LX 488-50, Coherent Inc.), 561 nm (OBIS, LX 561-50, Coherent Inc.), and 640 nm (OBIS, LX 640-100, Coherent Inc.). Laser lines were expanded, and coupled into a single beam using dichroic mirrors (427, 552, and 613 nm, LaserMUXTM, Semrock). An acousto-optic tunable filter (AOTFnc-400.650-TN, AA opto-electronics) was used as to modulate laser intensity. Light was circularly polarized using an achromatic quarter-wave plate. Two achromatic lenses were used to expand the excitation laser and an additional dichroic mirror (zt405/488/561/638rpc, Chroma) to direct it toward the back focal plane of the objective. Fluorescence light was spectrally filtered with emission filters (ET525/50 m, ET600/50 m, and ET700/75 m, Chroma Technology) and imaged on an EMCCD camera (iXon × 3 DU-897, Andor Technologies). The microscope was equipped with a motorized stage (MS-2000, ASI) to translate the sample perpendicularly to the optical axis. To ensure the stability of the focus during the acquisition, a homemade autofocus system was built. A 785-nm laser beam (OBIS, LX 785-50, Coherent Inc.) was expanded twice and directed toward the objective lens by a dichroic mirror (z1064rdc-sp, Chroma). The reflected IR beam was redirected following the same path than the incident beam and guided to a CCD detector (Pixelfly, Cooke) by a polarized beam splitter cube (PBS). Camera, lasers, and filter wheel were controlled with software written in Labview[52].

For image acquisition, on average, 30,000 frames (per detection channel in two-color acquisitions) were recorded at a rate of 50 ms/frame. Continuous excitation and activation were employed for all fluorophores employed in this work with powers as follows: 1 kW cm$^{-2}$ at 641 nm (for AF647), 0.8–1.2 kW cm$^{-2}$ at 561 nm (for mEos2), and 0–0.1 kW cm$^{-2}$ at 405 nm for activation. The intensity of activation was progressively increased throughout the acquisition to ensure a constant amount of simultaneously emitting fluorophores within the labeled structures. These excitation powers were optimized to ensure single-molecule detection, despite the large nuclear density of epigenetic compartments. More technical details and the method used to ensure single-molecule detection were previously described[32, 53].

**Postprocessing and analysis of two-color dSTORM data.** Unless stated otherwise, all homemade software and routines were developed in Matlab. Before further processing, super-resolution image quality was quantitatively assessed by using NanoJ-SQUIRREL[54]. Next, single-molecule localizations were obtained by using multiple-target tracing (MTT)[55]. Localization coordinates were further processed using SMLM_2 C, a custom software written in Matlab[32]. Fluorescent beads were used to correct for drift and chromatic aberrations. Lateral drift was corrected with 5 ± 3-nm precision, with a method already described elsewhere[52]. Chromatic aberration correction was performed using well-established protocols[32, 56]. Samples with abnormal drift or lesser precision of drift or chromatic aberration correction were discarded. Clustering of localizations was performed using an algorithm that was previously described in Cattoni et al.[53]. Colocalization of single-molecule detections was performed by using a custom implementation of the coordinate-based colocalization (CBC) analysis[57] adapted for whole-cell automated analysis[32]. Three additional methods were employed as controls: pixel, Pearson, and Manders correlation. For the latter, two-color digital images were reconstructed from the localization using standard procedures[52] and then used to plot the correlation between pixel intensities (pixel correlation analysis), or to calculate the Pearson or Manders correlation coefficients[58, 59].

**Analysis of one-color dSTORM data.** Single-molecule localizations are converted into a Voronoi diagram using a modified version of the Voronoi tesselation algorithm of Levet et al.[33]. Compartment segmentation is directly calculated from

the Voronoi diagram using three steps. First, densities of each polygon are calculated as the inverse of their area. Densities are then thresholded using the general criteria of Levet et al.[33]. Using this criterion, in which the threshold is determined by the average localization density, a random distribution of localizations did not provide any segmented polygon. Finally, polygons that have a density higher than the threshold and that are touching each other are merged to define the compartment outline. Compartment sizes are obtained by interpolating each segmented compartment on a grid of 5-nm size and calculating their equivalent diameter using standard morphological operations. Probability density functions in compartment-size histograms are calculated such that the area of each bar is the relative number of observations and that the sum of the bar areas is equal to 1.

**Analysis of genome-wide data.** Chromatin states were defined according to the enrichment in the percentages of H3K4me3 and H3K27me3/PC, as described in Supplementary Fig. 1. Calculation of the genomic size distributions of H3K27me3 and H3K4me3 domains (Supplementary Fig. 6) was performed as follows: (1) ChIP-chip/seq computed peaks were downloaded from ModEncode (ftp://data.modencode.org/D.melanogaster/)[60]. Data sets used are described in Supplementary Table 5. (2) Peak positions and intensities were used to resample the data and produce a continuous signal as a function of genomic position. (3) This signal was thresholded with a threshold of 0.1 of the log of the maximum intensity signal, ensuring that even peaks with very low intensity were retained. (4) Domains were defined as continuous segments extending for more than 2 bp with nonzero intensity. (5) Domains that were closer than 1 kb were fused together. This procedure was robust to calculate domain size distributions above 3 kb (Supplementary Fig. 6). (6) Finally, we estimated physical domain sizes from their genomic length as follows. The size of each genomic domain in bp was converted into nanometers using the empirical power law that relates genomic sizes to physical distances (Fig. 2e, f). The parameters of the power law depended on chromatin type (active or repressed) and on cell type (S2, early or late embryo), and are shown in the insets of Fig. 2e, f. After repeating this process for all genomic domain sizes, we obtained the distribution of domain sizes in nm for a specific chromatin type and cell type (Fig. 4f, g).

Clustering of domains of different epigenetic marks was defined as the ratio between the number of clusters of sizes larger than 150 nm obtained from Chip-seq vs. microscopy imaging. Changes in this threshold did not affect our main conclusions.

**In situ Hi-C data processing and normalization.** Hi-C data were processed using an in-house pipeline based on TADbit[61]. First, the quality of the reads was checked using the *quality_plot()* function in TADbit, which is similar to the tests performed by the FastQC program with adaptations for Hi-C data sets. Next, the reads are mapped following a fragment-based strategy, as implemented in TADbit where each side of the sequenced read was mapped in full length to the reference genome (dm3). After this step, if a read was not uniquely mapped, we assumed that the read was chimeric due to ligation of several DNA fragments. We next searched for ligation sites, discarding those reads in which no ligation site was found. The remaining reads were split as often as ligation sites were found. Individual split read fragments were then mapped independently. Next, we used the TADbit-filtering module to remove noninformative contacts and to create contact matrices. From the resulting contact matrices, low-quality bins (those presenting low contact numbers) were removed, as implemented in TADbit's *filter_columns()* function. Next, the matrices were normalized using the ICE algorithm[62]. The normalization iterations stopped when the biases were diverting less than 10% of the previous values or a max of 10 iterations. Finally, all matrices were corrected to achieve an average content of one interaction per cell. All parameters in TADbit were kept at default values.

The resulting late embryo and S2 Hi-C interaction maps (at 10-kb resolution) of the different replicates for each experiment were highly correlated (correlation coefficients from genomic distances ranging from 10 kb to 20 Mb were 0.99 to 0.75 and 0.95 to 0.45, respectively) and thus were further merged into the final data sets with more than 282-million and 210-million valid pairs each (Supplementary Table 6).

**Data availability.** The Hi-C data reported in this study are available at the Gene Expression Omnibus (GEO) repository under accession code GSE104961. Computer code and other data that support the findings of this study are available from the corresponding author upon reasonable request.

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

## Acknowledgements

We thank Brian Beliveau, Hien Hoang, and Ting Wu for their help with oligopaints design. This research was supported by funding from the European Research Council under the 7th Framework Program (FP7/2010–2015, ERC grant agreement 260787 to M.N. and FP7/2007–2013, and ERC grant agreement 609989 to M.A.M.-R.). M.A.M.-R. and G.C. acknowledge support from the European Union's Horizon 2020 research and innovation program under grant agreement 676556. This work has also benefited from support by the Labex EpiGenMed, an «Investments for the future» program, reference ANR-10-LABX-12-01, the Spanish Ministry of Economy and Competitiveness (BFU2013-47756-P to M.A.M.-R.), and from "Centro de Excelencia Severo Ochoa 2013–2017", SEV-2012-0208 to the CRG. 3D-SIM experiments were performed at Montpellier Resource Imaging. We acknowledge the France-BioImaging infrastructure supported by the French National Research Agency (ANR-10-INBS-04, «Investments for the future»).

## Author contributions

D.I.C, A.M.C.G. M.G. and M.N. designed the experiments and conducted the research. A.V. J.-B.F., M.G. and M.N. developed the software for image analysis. D.I.C, A.M.C.G., A.V. and F.B. designed Oligopaint probes. F.B. performed fly handling. M.D. and M.A.M.-R. performed Hi-C bioinformatics analysis. D.C., C.H. and S.D. synthesized and purified oligopaint libraries and performed S2 cells handling. D.I.C, A.M.C.G. M.G., G.C. and M.N. wrote the manuscript. All the authors reviewed and commented the data.

## Additional information

**Competing interests:** The authors declare no competing financial interests.

