## [Peer Review File · Nature Communications]

Reviewers' comments:

Reviewer #1 (Remarks to the Author):

In their manuscript Cattoni et al. combine super-resolution single-cell imaging and correlative analysis of genome-wide next-generation sequencing (ChIP-seq and HiC) data to study variability of multi-level chromatin domain (TAD) organization in different *Drosophila* cell types. Using fluorescence in situ hybridization (FISH) detection of an impressive set of oligo-paint probes covering TAD boundary regions as well as inter-TAD sequences, they observe (not unsurprisingly) a high degree of stochasticity, with co-localization of neighboring TAD boundaries being less frequent than commonly implied by HiC data, therefore 'painting' a more dynamic picture of TAD organisation. Using dSTORM imaging of immuno-fluorescently labelled active and repressive histone marks the authors furthermore demonstrate a non-homogeneous distribution of both marks with cell-type specifically modulated clustering.

Overall, this study is very well executed, thoroughly controlled, and the conclusions drawn are largely sound. To my knowledge, this is one of the first studies that uses cutting-edge imaging to measure single-cell contact probabilities and combine this with NGS methods in such comprehensive way. This work extends and complements previous NGS based studies and will significantly contribute to our understanding of modular genome organization of higher eukaryotes. It will be important (albeit not within the scope of this manuscript) to see how far the results from *Drosophila* are transferable to mammalian genomes, featuring differences in TAD sizes and the employment of cohesin/CTCF.

Before recommending this manuscript unreservedly for publication in Nature Communications I would like to see the following points addressed:

Major issues:

- I) One strength of this paper is the usage of complementary super-resolution imaging techniques. While SIM imaging seems well-executed and the data analysis straight forward and credible, I feel this is less so the case for dSTORM imaging (here citations are not well balanced either, with the latter referring to 3 citations vs none for SIM).
 - 1.) While quality control of SIM data using SIMcheck is very commendable (the primary paper could be cited for the reader's information!), the same should be done with dSTORM data using the novel NanoJ SQUIRREL ImageJ/Fiji plugin.
 - 2.) The authors apparently apply dSTORM without oblique (HiLo) illumination or 3D-localization. How is that affecting the validity of the results? Related to this, dSTORM imaging of dense nuclear labeling is notoriously difficult as blinking events may not be properly separated, which may lead to local artifacts, such as merging of structures. The latter could affect the analysis of epigenetic domain clusters.
 - 3.) Blinking properties and isolation of blinking events depends on the excitation strength, the authors should indicate the laser powers used (in mW/cm²). What other acquisition parameter were used?
 - 4.) As dSTORM image reconstructions are dependent on the algorithms employed, the authors may need to comment on why they used custom scripts for localization rather than more common software, such as RapidSTORM.
 - 5.) Manders' and Pearson's coefficients seem less well-suited approach to analyze spatial relationships in STORM data. A positive control for truly co-localizing targets is also missing.
- II) The classification of TAD chromatin states, as outlined in Fig S2 a,b, is somewhat oversimplified and ambiguous. Many regions are assigned 'active' even though they are significantly enriched for silencing marks, and more TADs seem to have 'mixed' signatures (unlike the rather untypically

extreme examples shown in panel a). Particular in the 'late embryo' stage TADs seem to be assigned rather arbitrarily to one of the three classes despite of almost identical amount of active and silencing mark(s). Accordingly, the color scale bar is rather misleading as it implies TADs having either one or the other. Also, negative values are somewhat nonsensical, as well as the annotation "enrichment" (over which baseline?). Instead two bars, each for active and silencing marks, should be displayed and annotated with "fraction" or "relative amount".

Minor issues:

p8 (top): "...discrete nanometer-sized compartments..." might be misleading as it implies a range of 1-10 nm, although its rather in the 10s to 100s nm range.

Fig. 1d, Fig. 3a,b and others: In terms of developmental stages, should "early embryo" not be displayed before "late embryo" and "S2" stages? What is the "developmental state" of S2 cells compared to embryo cells?

It is unclear, whether the authors generated the HiC data themselves or relied on public databases. Please specify the protocol (if applicable) in the methods section.

Typos and editing:

p9 (bottom): "In this work, we showed that genome in *Drosophila*..."

p25: Several expression errors, whole paragraph needs revision! The method of how to calculate physical domain sizes from genomic data does not get clear.

Fig. 1a (legend): Indicate whether the image shows a projection. Maybe consider showing a lateral and axial view. Furthermore, does the scale bar size of 1 μm annotate the overview or the inset magnification?

Fig. 1e/2a: y-axis, what is the unit of contact frequency? Relative, absolute, arbitrary units?

Fig. 1f: active triangle is hardly visible.

Fig. 2e/f: What is the unit of "Norm. physical distance"? The difference between "Physical distance, nm" in panel b and "Norm. physical (end-to-end) distance" should be clarified. X- and y-axis scaling should be identical in e and f.

Fig. 3c (legend): Solid grey line represent the chromatin fiber (instead of DNA fiber)?

Fig. S1i: The higher standard deviation of distance measurements in late embryos indicates a larger variability, but does this also directly indicate higher dynamics? What means the expression "as obtained from bootstrapping"?

Fig. S2b,c: dark blue and black circles are hard to distinguish. For the annotation of colors and symbols the legend of S2c refers to Fig. S2b, which then refers to Fig. S1 and Fig. 2b. It might be easier to simply repeat the annotation.

Fig. S3c: Principle component analysis requires more explanation, e.g. of the color code.

Fig. S4c(ii). Box is wrongly positioned.

Fig. S4d: The diagram of the statistical evaluation is missing

Extended data, p4: whole paragraph is very sloppily edited and requires revision.

Extended data, p15: values in table should be indicated in μm (not nm), to avoid false impression of nm-accuracy.

Generally, all figure annotations need to be revised, e.g. only first letters should be capitalized and font type/sizes unified.

Reviewer #2 (Remarks to the Author):

Cattoni et al. use super-resolution imaging and Oligopaints-based probes to visualize multiple pairs of regions on two *Drosophila* chromosomes to probe the higher-order organization of topologically associated domains (TADs) in single cells. Using probes targeting multiple TAD borders, they find that interactions between regions occur relatively rarely ($\sim 10\%$ of the time) and are governed by considerable variation and stochasticity. They further specify that chromatin state, cell type, and locus-specific interactions act to modulate that stochasticity.

The data are technically sound and the findings are interesting as they relate single cell data to population-based Hi-C data, which is much needed in the field. The study carries implications for the understanding of how genome architecture acts throughout development and gene regulation.

We have no major concerns with this study, but a few minor points to further improve the manuscript:

1. While the authors provide modeling experiments to demonstrate that a 50% reduction in volume is not sufficient to explain the differences in density observed between S2 cells and embryonic cells, an analysis on whether the number of clusters or median distance between clusters in Figure 3 correlates with the size of the cell nucleus within a cell type would further validate that the observed differences are not due to cell size.
2. Does an exponential model, or a power-law model, best fit the data comparing Hi-C contact frequency to absolute contact probability?
3. While the differences between S2 and embryonic cells are striking, it is worth noting that these differences could be between cell types or between primary and immortalized cells. In fact, late stage and early stage embryonic samples appear more similar than they are different; without another cell line or another primary tissue it is difficult to say that this is truly a developmental difference and not amplified by the process of immortalization and cell culture.
4. What is the coefficient of variation for distance between two regions? How does it vary between cell types? Since standard deviation scales linearly with mean distance (Fig S1i), it is likely to be constant, and would be a useful quantifier for the observed stochasticity.

Overall, we think this is a well-done and interesting study with highly suggestive implications for the field.

Reviewer #3 (Remarks to the Author):

In this manuscript, Cattoni and colleagues attempt to compare physical distances between certain regions of the genome in the nucleus using super-resolution imaging, coupled to fluorescence in situ

hybridization and immunofluorescence, with high-resolution Hi-C data, in *Drosophila* embryos at two developmental stages (early/stage 5 and late/stage 16) and in cultured S2 cells.

They labelled TAD boundaries using FISH and imaged cells using 3D-SIM to assess the variability in TAD organization between single cells. While this is a valid and an interesting approach, there are some points that need to be better defined, discussed or analysed. In particular, the discussion of models of TAD formation is incomplete. First of all, the most popular and current loop extrusion model for TAD formation (at least in mammalian cells) is not discussed. This is important because this model does not require stable looping of TAD boundaries – a central assumption of this manuscript – instead TADs arise from a dynamic balance between cohesion-mediated loop extrusion, the blocking of that movement by architectural proteins and activities that either promote loop extrusion by cohesion (Nipbl) or that remove cohesin from the chromosome (WAPL). Are the authors' data consistent with this model? A model of TAD boundaries in the fly genome, based on simple nucleosome properties – especially at active genes (Uljanov et al, *Genome Research*, 2016) is also not discussed.

The authors place a lot of emphasis on the stochastic nature of chromatin organization, without a proper citation of the well-studied levels of stochastic genome organization in the literature (e.g. association with the nuclear lamina, chromosome territory neighborhoods, gene positioning within chromosome territories).

More specific points:

1. Are the probes located inside the TAD (adjacent to its boundary), overlapping the TAD boundary, or just outside (in the inter-TAD or the adjacent TAD)?
2. In Fig. 1C the boxed legend is labeled incorrectly. Presumably filled circles should be TB2-TB3, not TB2-TB2
3. In Fig. 2D, it is more common to show this kind of plot in log-log scale (with log spaced bins on x axis), this way power-law dependencies look like straight lines and their slope can be more easily estimated.
4. In Fig. 2b and S2b it is unclear what was taken as the physical distance – was it the mean of the distance between a pair of probes? Then, perhaps, error bars for this measurement need to be shown too?
5. It would be interesting to directly compare (and discuss differences, if there are any) power-law scaling exponents between Hi-C data and FISH data, since analysis of contact probability decay is a common way to assay polymer properties of the genome from Hi-C data. And since the authors have access to physical distance measurements between pairs of loci at many separations, a discussion of chromatin polymer properties based on this would be warranted – i. e. whether the resultant curve is consistent with existing models of chromatin organization.
6. In Fig. 3b, after labelling 69 foci on chromosome 3R the authors obtained an average of 89 spots in S2 cells, a lot more than expected – is it perhaps due to them having 4 copies of each chromosome, or chromosome rearrangements in the population? The latter could also explain a somewhat bi-modal distribution of distances in Fig. 3B if the rearrangements are only present in a part of the population, but both explanations unfortunately would invalidate the whole comparison of this cell line to embryos with this approach.
7. In the STORM analysis of compartment clustering, how does the proportion of clusters unaccounted for from modelling based on 1D distribution of histone marks correspond to frequency of interactions between the corresponding regions by Hi-C? One would expect to see higher interaction frequencies between K27me3 regions in the data from embryos than from S2 cells (and the same for K4me3), and this would be a good validation of the approach. And a similar question can be asked about compartment densities (Fig. S6a) – do FISH measurements correlate with Hi-C data, e.g. when comparing number of contacts inside K27me3 and K4me3 domains?

Reviewers' comments:

Reviewer #1:

General comment

In their manuscript Cattoni et al. combine super-resolution single-cell imaging and correlative analysis of genome-wide next-generation sequencing (ChIP-seq and HiC) data to study variability of multi-level chromatin domain (TAD) organization in different Drosophila cell types. Using fluorescence in situ hybridization (FISH) detection of an impressive set of oligo-paint probes covering TAD boundary regions as well as inter-TAD sequences, they observe (not unsurprisingly) a high degree of stochasticity, with co-localization of neighboring TAD boundaries being less frequent than commonly implied by HiC data, therefor 'painting' a more dynamic picture of TAD organisation. Using dSTORM imaging of immuno-fluorescently labelled active and repressive histone marks the authors furthermore demonstrate a non-homogeneous distribution of both marks with cell-type specifically modulated clustering.

Overall, this study is very well executed, thoroughly controlled, and the conclusions drawn are largely sound. To my knowledge, this is one of the first studies that uses cutting-edge imaging to measure single-cell contact probabilities and combine this with NGS methods in such comprehensive way. This work extends and complements previous NGS based studies and will significantly contribute to our understanding of modular genome organization of higher eukaryotes. It will be important (albeit not within the scope of this manuscript) to see how far the results from Drosophila are transferable to mammalian genomes, featuring differences in TAD sizes and the employment of cohesin/CTCF.

Answer:

We thank Reviewer #1 for his/her thorough and careful reading of the manuscript and his/her positive criticism. We explain how we addressed the specific comments and suggestions below.

Specific comments

Before recommending this manuscript unreservedly for publication in Nature Communications I would like to see the following points addressed:

1) One strength of this paper is the usage of complementary super-resolution imaging techniques. While SIM imaging seems well-executed and the data analysis straightforward and credible, I feel this is less so the case for dSTORM imaging (here citations are not well balanced either, with the latter referring to 3 citations vs none for SIM).

Answer:

We apologize for the lack of citations when referring to 3D-SIM microscopy. In the revised version of the manuscript we have included the seminal papers referring to 3D-SIM listed below (page 5):

Surpassing the lateral resolution limit by a factor of two using structured illumination microscopy
Gustafsson MG, *J. Microsc.*, 2000

Three-dimensional resolution doubling in wide-field fluorescence microscopy by structured illumination, Gustafsson MG, Shao L, Carlton PM, Wang CJ, Golubovskaya IN, Cande WZ et al. *Biophys. J.*, 2008

1.) *While quality control of SIM data using SIMcheck is very commendable (the primary paper could be cited for the reader's information!), the same should be done with dSTORM data using the novel NanoJ SQUIRREL ImageJ/Fiji plugin.*

Answer:

We apologize for the lack of the citation of the SIMcheck software, this has been corrected in the revised version (page 19 in the Material and Methods section).

For all our experiments in dSTORM image quality was evaluated by visually inspecting the wide field images and super-resolution rendering and by systematically evaluating fluorescent traces to ensure single molecule imaging conditions and optimization of fluorophore photophysics (see also answer to comment 2). Following the reviewer's advice, we have employed the NanoJ-SQUIRREL software (Culley et al. 2017) to further check the quality of our dSTORM images. Below are depicted representative images of wide-field, dSTORM reconstructions and error maps for S2 cells bearing H3K27me3 labelled with AF647. Equivalent analysis were performed for all cellular types and different epigenetic marks labelled with different fluorophores. In all cases the error maps confirmed that our dSTORM conditions did not introduce large scale image artifacts and that super resolution images faithfully recovered at the nanoscale the structural features that could be inferred from the low resolution wide field images. These new controls are now cited in the main text (page 22 Material and Methods section).

Image quality analysis with NanoJ-SQUIRREL

Representative wide-field, dSTORM and error map images of H3K27me3-AF647 in S2 cells nuclei. Error maps were obtained by employing NanoJ-SQUIRREL with standard parametrization. Resolution scaling function (RSF) was optimized and maximum magnification was set to 5. Similarity between diffraction limited images and the images obtained from applying the RSF to the super resolution image was evaluated by using Resolution Scaled Error (RSE). In all cell types and epigenetic marks, the RSE values were globally lower than 150.

2.) The authors apparently apply dSTORM without oblique (HiLo) illumination or 3D-localization. How is that affecting the validity of the results? Related to this, dSTORM imaging of dense nuclear labeling is notoriously difficult as blinking events may not be properly separated, which may lead to local artifacts, such as merging of structures. The latter could affect the analysis of epigenetic domain clusters.

Answer:

In our initial optimization process, we compared epifluorescence with HiLo illumination. Because the typical excitation width of HiLo (6-8 μm) (Tokunaga et al. 2008) is comparable to the diameter of *Drosophila* nuclei ($\sim 5 \mu\text{m}$), we did not observe an important difference between these illumination modes in our conditions.

We have also made acquisitions using 3D-localization (astigmatism). In our hands, this method had, however, a number of drawbacks: (1) axial localization precision degraded rapidly with distance to focal plane. This led to a deformation of the 3D density of domains in the axial direction; (2) localizations away from the focal plane displayed lower signal to noise ratios, therefore are more likely to be missed by the localization software. This meant that the density in 3D reconstructions was strongly modulated by axial position; (3) the best depth of field possible under conditions was only $\sim 600\text{-}800\text{ nm}$, representing less than one fifth of the nucleus volume; (4) epigenetic domains that are only partly present within this narrow depth of field were severed, thus leading to an underestimation of their actual size. Thus, because these drawbacks of 3D-localization methods could have led to similar or worse biases in the analysis of epigenetic domains, we decided to use 2D-SMLM. We envision that future experiments could profit from new technical developments that would permit efficient 3D localization in very thick specimens ($>5\mu\text{m}$) under realistic biological conditions.

Our approach, also used by leaders in the field (Boettiger et al. 2016; Ricci et al. 2015), was to use 2D-localization to estimate the size distributions of epigenetic domains. In our implementation, we obtained a depth of field of $\sim 500\text{ nm}$. Thus, our 2D super-resolved images are a 2D projection of the 3D density of epigenetic domains. As the large majority of domains detected ($>95\%$) were smaller than the depth of field, then we would only expect to slightly underestimate the size of large domains. Our main conclusion from these measurements was that epigenetic size distributions were not large enough to account for the large epigenetic compartments observed by STORM. Thus, we think our biological conclusion is still sound, despite a possible underestimation of the degree of clustering of epigenetic domains. In the revised manuscript, we now discuss these issues in the legend of Supplementary Fig. 6.

Finally, we agree that dSTORM imaging of dense nuclear labeling is difficult due to overlap of blinking events. To overcome this problem, we did an extensive optimization and validation of our protocol (Georgieva et al. 2014). We observed that using high excitation densities and low photoactivation powers is critical to ensure that we detect exclusively single-molecules (i.e. detection of two molecules within a distance close to the PSF is extremely rare or impossible). The description of the full protocol used and the verifications made to ensure single-molecule conditions were extensively described in two methodological papers that we now cite (Georgieva et al. 2014; Cattoni et al. 2013). In the revised manuscript, we now mention the criteria used to ensure single-molecule detection conditions (Materials and Methods section, page 21).

'These excitation powers were optimized to ensure single-molecule detection, despite the large nuclear density of epigenetic compartments. More technical details and the method used to ensure single-molecule detection are described elsewhere (Georgieva et al. 2016; Cattoni et al. 2013).'

3.) Blinking properties and isolation of blinking events depends on the excitation strength, the authors should indicate the laser powers used (in mW/cm²). What other acquisition parameters were used?

Answer:

We apologize to the reviewer for the absence of this information in the material and methods section. The information has been added to the Methods section (page 21) and now it reads:

'For image acquisition, on average 30,000 frames (per detection channel in two-color acquisitions) were recorded at a rate of 50 ms/frame. Continuous excitation and activation was employed for all fluorophores employed in this work with powers as follows: 1 kW/cm² at 641 nm (for AF647), 0.8–1.2 kW/cm² at 561 nm (for Cy3b), and 0–0.1 kW/cm² at 405 nm for activation. The intensity of activation was progressively increased throughout the acquisition to ensure a constant amount of simultaneously emitting fluorophores within the labeled structures.'

4.) As dSTORM image reconstructions are dependent on the algorithms employed, the authors may need to comment on why they used custom scripts for localization rather than more common software, such as RapidSTORM.

Answer:

We apologize to the reviewer if the information regarding the localization algorithm was not clearly stated in the methods section. We did not employ custom-made scripts for single molecule localization in dSTORM imaging, instead we used the Multiple Target Tracing (MTT) a software package developed in 2008 by Marguet's group (Sergé et al. 2008) and extensively used before by our group (Marbouty et al. 2015; Le Gall et al. 2016; Fiche et al. 2013; Faure et al. 2016). MTT was and is broadly used by the community due to its robustness and high reliability in single-molecule detection.

5.) Manders' and Pearson's coefficients seem less well-suited approach to analyze spatial relationships in STORM data. A positive control for truly co-localizing targets is also missing.

Answer:

In the original version of the manuscript, we employed four methods to analyze the colocalization between epigenetic marks: single-molecule coordinate-based colocalization (CBC) analysis (Tarancón Díez et al. 2014) adapted for whole-cell automated analysis (Georgieva et al. 2014), as well as pixel, Pearson's, and Manders's correlation. From all of these methods, we could conclude that there is marginal co-localization between H3K27me3 and H3K4me3 compartments.

We agree with the reviewer in that Manders' and Pearson's coefficients are less well-suited for super-resolution datasets, while CBC has been specifically designed and validated for single-molecule-based super-resolution (Tarancón Díez et al. 2014; Georgieva et al. 2014). Thus, instead of showing Manders' co-localization in the main figures and CBC analysis in the Supplement, we now show CBC analysis in Fig. 4b and the more traditional co-localization analysis methods in the Extended data (Supplementary Fig. 4ciii). The revised version of the text now reads (page 11):

'Repressed and active chromatin marks were strictly segregated at the nanoscale for all cell types, as revealed by coordinate-based co-localization analysis (aCBC, (Georgieva et al. 2016), Fig. 4b). These findings were confirmed by independent colocalization methods and by additional controls using doubly-labeled nuclear factor and non-colocalizing epigenetic marks (Supplementary Figs. 4b-d).'

The legend of Fig. 4b now reads:

'Quantification of co-occurrence ($C_A > 0.5$) between active and repressive chromatin using aCBC (Georgieva et al. 2016). Boxplots of CAs for H3K4me3 and H3K27me3 are shown in the upper panel and lower panels, respectively.'

Next, the reviewer requests a positive control for colocalization. To answer this question, we now show 2-color dSTORM imaging of S2 cells labeled with Beaf-32-mEos2 and immunostained with a primary Beaf-32 antibody coupled to AF647. Beaf-32 is an insulator protein that binds extensively to TAD barriers (Van Bortle & Corces 2013; Sexton et al. 2012). These experiments were performed using identical imaging conditions to those employed to study epigenetic compartments by 2-color dSTORM (Fig. 4). Co-localization-based single-molecule images show a clear positive co-localization between Beaf-32-mEos2 and Beaf-ab-AF647 (see below and new Supplementary Fig. 4b-i). Analysis of the CBC coefficient from many cells (N=14) show colocalization coefficients between 0.7-0.8 (see below and Supplementary Fig. 4b-ii), in contrast to CBC coefficients obtained for K27me3-K4me3 (0.1-0.2, Fig. 4b).

Supplementary figure 4 panel b i-ii: (i) Two-color SMLM imaging of Beaf-32. Beaf-32 was labeled directly using a fusion protein (Beaf-32-mEos2) and by immunofluorescence (primary Beaf-32 antibody conjugated to AF647). Cells were then imaged sequentially in these two channels and analysed using the aCBC analysis. Left panel shows the aCBC map of the AF647 channel, and the right panel displays the aCBC map of the mEos2 channel. The numbers (N) at the bottom of each cell correspond to the number of single detections in each map obtained from 20,000 frames and are an indication of the density of events. The color scale on the right

(Colocalization values) reflects the aCBC coefficient for each single localization. Values above 0.5 indicate a high degree of colocalization. N=14. Scale bars: 1 μ m (ii) Statistics of the colocalization for the Beaf-32-AF647 and Beaf-32-mEos2 channels. Boxplots indicate the median (orange bar), 25th and 75th percentile limits (blue areas), and extreme values (whiskers) of the fraction of events with aCBC colocalization value (C_A) > 0.5.

6.) The classification of TAD chromatin states, as outlined in Fig S2 a,b, is somewhat oversimplified and ambiguous. Many regions are assigned 'active' even though they are significantly enriched for silencing marks, and more TADs seem to have 'mixed' signatures (unlike the rather untypically extreme examples shown in panel a). Particular in the 'late embryo' stage TADs seem to be assigned rather arbitrarily to one of the three classes despite of almost identical amount of active and silencing mark(s). Accordingly, the color scale bar is rather misleading as it implies TADs having either one or the other. Also, negative values are somewhat nonsensical, as well as the annotation "enrichment" (over which baseline?). Instead two bars, each for active and silencing marks, should be displayed and annotated with "fraction" or "relative amount".

Answer:

We agree with the reviewer that the representation chosen for the Supplementary Fig. 2a-b was confusing. We apologize also for the negative values for fraction of enrichment of H3K27me3 and PC appearing in the previous version of the figure, this was a typo due to figure generation parameters. Following the reviewers advice we have modified the figure and now a single bar with a single color code displays the relative amount of each mark in each TAD. We have conserved the color code (red, blue, black) for the resulting TAD state displayed in the lower panel. For panel a and also to avoid confusions, we have decided to replace the colors by a grey scale that is attributed following the criteria described in Supplementary Fig. 1. Finally, we agree with the reviewer that assignation of TAD state in some cases is rather difficult, particularly when both active and repressed marks are present in similar proportions. In all cases where the proportions between H3K4me3 and H3K27me3 were similar and significant (above 20-25 % relative amount), the presence of polycomb proteins was employed to define whether or not a TAD was repressed. In its presence, the TAD was defined as repressed, in its absence it was defined as inactive (solely two TADs were classified as inactive according to this definition, B1-2 and B9-10 in early embryo).

Minor issues:

7.) *p8 (top): "...discrete nanometer-sized compartments..." might be misleading as it implies a range of 1-10 nm, although its rather in the 10s to 100s nm range.*

Answer:

We agree with the reviewer's comment. We have modified the text and now it reads (page 10):

'...revealed that active and repressive histone marks distributed non-homogeneously across the cell nucleus, forming discrete compartments of tens to hundreds of nanometers for all cell types (Figs. 4a and Supplementary Fig. 4a).'

8.) *Fig. 1d, Fig. 3a,b and others: In terms of developmental stages, should "early embryo" not be displayed before "late embryo" and "S2" stages? What is the "developmental state" of S2 cells compared to embryo cells?*

Answer:

We agree with the reviewer comment that intuitively the order of presentation of panels should follow the developmental stages. However, we have rather chosen to display the panels in the order that is defined by the experimental findings. In particular, we wished to highlight how the levels of stochasticity are modulated between cell-types. We found that the clearest way to reinforce this message for the reader was to present our results ordered from highest to lowest in terms of probability of interaction or structural compaction at different chromosomal scales.

The S2 cell line was derived from a primary culture of late stage (20-24 hours old) embryos (Schneider 1972) but displays important genomic and expression pattern changes with respect to cell in this developmental state. Thus, S2 cells are not considered similar to late embryonic cells.

9.) *It is unclear, whether the authors generated the HiC data themselves or relied on public databases. Please specify the protocol (if applicable) in the methods section.*

Answer:

We apologize to the reviewer for the lack of clarity regarding the Hi-C data employed in our work. Late embryo Hi-C data is from Sexton *et al.* (Sexton *et al.* 2012) and S2 data was generated specifically for this work. This last dataset is being currently deposited. The references to these datasets will be included in the Materials and Methods section of the final version.

Typos and editing:

10.) p9 (bottom): *“In this work, we showed that genome in Drosophila...”*

Answer:

The extra space has been removed.

11.) p25: *Several expression errors, whole paragraph needs revision! The method of how to calculate physical domain sizes from genomic data does not get clear.*

Answer:

We apologize for the lack of clarity. We have rewritten this paragraph and added details to the method to calculate physical domain sizes from genomic data to improve clarity and readability. The revised version of the methods section now reads (pages 23-24):

*‘Calculation of the genomic size distributions of H3K27me3 and H3K4me3 domains (Supplementary Fig. 6) was performed as follows: (1) ChIP-chip/seq computed peaks were downloaded from ModEncode (<ftp://data.modencode.org/D.melanogaster/>) (Contrino *et al.* 2012). Datasets used are described in the Online Methods (Supplementary Table 5). (2) Peak positions and intensities were used to resample the data and produce a continuous signal as a function of genomic position. (3) This signal was thresholded with a threshold of 0.1 of the log of the maximum intensity signal, ensuring that even peaks with very low intensity were retained. (4) Domains were defined as continuous segments extending for more than 2 bp with non-zero intensity. (5) Domains that were closer than 1 kb were fused together. This procedure was robust to calculate domain size distributions above 3kb (Supplementary Fig. 6). (6) Finally, we estimated physical domain sizes from their genomic length as follows. The size of each genomic domain in bp was converted into nanometers using the empirical power law that relates genomic sizes to physical distances (Figs. 2e-f). The parameters of the power-law depended on chromatin type (active or repressed) and on cell type (S2, early or late embryo), and are shown in the insets of Figs. 2e-f. After repeating this process for all genomic domain sizes, we obtained the distribution of domains sizes in nm for a specific chromatin type and cell type (Fig. 4f-g).’*

12.) Fig. 1a (legend): *Indicate whether the image shows a projection. Maybe consider showing a lateral and axial view. Furthermore, does the scale bar size of 1 μ m annotate the overview or*

the inset magnification?

Answer:

The legend of Figure 1a now indicates that the image depicted is three dimensional and the scale of the main image and inset are indicated in the legend to comply with the journal's editorial guidelines. Following the reviewer's advice we have added a rotated view and inset of the same cell. The new figure is shown below.

a

Figure 1a. Top, region of Hi-C contact matrix of chromosome 2L. Black dotted line demarcates a single TAD and pink and cyan boxes represent the TAD borders (TB) labelled by Oligopaint. Chromatin epigenetic state is indicated at the bottom using the color code of panel b. Bottom, representative three-color 3D-SIM image in two different orientations. DAPI is shown in gray, TB2 in pink and TB3 in cyan. Scalebar = 1 μ m for main image. 5x amplification of selected region is depicted in inset.

13.) Fig. 1e/2a: y-axis, what is the unit of contact frequency? Relative, absolute, arbitrary units?

Answer:

We apologize for the lack of clarity in the units of Hi-C data quantifications. The Hi-C contact frequency is expressed throughout the paper as “normalized Hi-C counts”. We have now modified the y-axis of Figs 1e, 2a, 2d and S2a, and modified the legends accordingly.

14.) Fig. 1f: active triangle is hardly visible.

Answer:

We have changed the color tone and enhanced the contrast of the triangle to improve its visibility.

15.) Fig. 2e/f: What is the unit of “Norm. physical distance”? The difference between “Physical distance, nm” in panel b and “Norm. physical (end-to-end) distance” should be clarified. X- and y-axis scaling should be identical in e and f.

Answer:

We apologize for the lack of clarity in our definitions of axes labels and scaling. In the revised version of the figure legend, we indicate that normalized data was obtained by calculating the ratio between the mean physical distances and the best fitting values of the pre-exponential terms obtained from the power law fitting to the raw data depicted in figure S2b. Additionally, to avoid any confusion, the expression ‘end-to-end’ was replaced by ‘mean’. Following the reviewers’ advice, the X- and Y-axis scales of all panels have been optimized to improve visibility and ease the comparison between panels.

In the revised version of the manuscript, the legend of Figs. 2e-f now reads:

‘Log-log plot of the mean physical distance vs. genomic length for (e) active and (f) inactive/repressed chromatin domains for different cell types. Mean distance values were normalized by the pre-exponential factor from the power-law fit of each dataset (Supplementary Figs. 2d-e). Solid lines show the power-law fits, with the scaling exponents β shown as insets. Circles and triangles are depicted as described in panel 2b. Error bars represent SEM. $N > 140$ for each data point, from more than three biological replicates (see Supplementary Fig. 1).’

16.) Fig. 3c (legend): Solid grey line represent the chromatin fiber (instead of DNA fiber)?

Answer:

In the legend of figure 3c ‘DNA’ has been replaced by ‘chromatin’

17.) Fig. S1i: The higher standard deviation of distance measurements in late embryos indicates a larger variability, but does this also directly indicate higher dynamics? What means the expression “as obtained from bootstrapping”?

Answer:

We agree with the reviewer that higher standard deviation of distance measurements in late embryos indicates a larger variability but not necessarily higher dynamics. We have modified the legend of Supplementary Fig. 1i accordingly:

‘...indicating that for equivalent mean size of TADs their structure displays higher variability in late embryonic cells.’

To obtain the standard error on each point we employed the bootstrapping method. This technique involves the random resampling of the original dataset into subsets (called a ‘resample’ or bootstrap sample) with replacement. Each observation is chosen randomly and entirely by chance, such that each observation has the same probability of being chosen at any

stage during the resampling process. Because of this, for all practical purposes there is virtually zero probability that the 'resample' will be identical to the initial sample. We repeated this process a 100 times for a dataset of distances, and for each of these 'resamples' we compute the standard deviation and the contact probability, called bootstrap estimates. From the standard deviation of these 100 estimates, we then obtain the errors of the each of the variables.

This procedure has now been better described in the extended data, which now reads:

'Error bars represent the standard error of the mean (SEM) as obtained from "bootstrapping", by randomly resampling with replacement each dataset one hundred times to estimate the errors in standard deviations and contact probabilities.'

18.) Fig. S2b,c: dark blue and black circles are hard to distinguish. For the annotation of colors and symbols the legend of S2c refers to Fig. S2b, which then refers to Fig. S1 and Fig.2b. It might be easier to simply repeat the annotation.

Answer:

We agree with the reviewer that the annotation of colors was perhaps not clear in the legend. The legend of Supplementary Fig. 2b now reads:

"Chromatin state of domains encompassed by borders is color-coded as follows, red: active, blue: repressed, black: inactive."

19.) Fig. S3c: Principle component analysis requires more explanation, e.g. of the color code.

Answer:

We apologize for the omission. We have now included the following explanation:

"Principal Component Analysis (PCA) of the normalized $p(r)$ distributions from Fig. 3b, displaying the first two principal components scores of every cell. Color-code indicates the kernel density estimation heatmap, the density is calculated on the number of points in a location, with larger number of clustered points resulting in larger values. Blue: low, yellow: high."

20.) Fig. S4c(ii). Box is wrongly positioned.

Answer:

We corrected the position of the box in the revised version of Supplementary Fig. 4.

21.) Fig. S4d: The diagram of the statistical evaluation is missing

Answer:

To address this remark, we have added the compartment size distributions from single-color dSTORM imaging of H3K4me3 labelled with antibodies bearing either Cy3b or Alexa 647 (see below).

New Supplementary Figure 4e. (i-ii) Representative images of single-color dSTORM imaging of H3K4me3 labelled with antibodies bearing different organic fluorophores (either Cy3b or Alexa 647, N=35). Scalebar = 500 nm. (iii) Distribution of H3K4me3 compartment sizes for either AF647 or Cy3b. From these experiments, we conclude that no significant differences in the spatial localization, and distribution of sizes of compartments were observed when Cy3b was used instead of AF647.

22.) *Extended data, p4: whole paragraph is very sloppily edited and requires revision.*

Answer:

We apologize for the lack of clarity in the description of oligopaint related protocols. The whole section has been thoroughly revised and shortened significantly to improve its clarity.

23.) *Extended data, p15: values in table should be indicated in μm (not nm), to avoid false impression of nm-accuracy.*

Answer:

The units of the values of Table have been amended and are displayed now in μm .

24.) *Generally, all figure annotations need to be revised, e.g. only first letters should be capitalized and font type/sizes unified.*

All figures annotations have been thoroughly revised and modified accordingly to the reviewer's suggestions and journal formatting requirements.

Reviewer #2:

General comment

Cattoni et al. use super-resolution imaging and Oligopaints-based probes to visualize multiple pairs of regions on two Drosophila chromosomes to probe the higher-order organization of topologically associated domains (TADs) in single cells. Using probes targeting multiple TAD borders, they find that interactions between regions occur relatively rarely (~10% of the time) and are governed by considerable variation and stochasticity. They further specify that chromatin state, cell type, and locus-specific interactions act to modulate that stochasticity. The data are technically sound and the findings are interesting as they relate single cell data to population-based Hi-C data, which is much needed in the field. The study carries implications for the understanding of how genome architecture acts throughout development and gene regulation.

We thank the reviewer for her/his careful reading of our manuscript and for her/his comments and suggestions. We have taken them into account and included all modifications in the revised version of the text. A detailed answer to each of the comments is given below.

Minor issues:

We have no major concerns with this study, but a few minor points to further improve the manuscript:

1.) While the authors provide modeling experiments to demonstrate that a 50% reduction in volume is not sufficient to explain the differences in density observed between S2 cells and embryonic cells, an analysis on whether the number of clusters or median distance between clusters in Figure 3 correlates with the size of the cell nucleus within a cell type would further validate that the observed differences are not due to cell size.

Answer:

We find the reviewer's suggestion very pertinent. We agree that an additional control to evaluate that changes in the nucleus size are not correlated with the observed changes in the number of detected clusters is important. Following the reviewer's advice we have re-analyzed our data for each cell type and evaluated the dependence of the number of clusters detected vs. nucleus size (see figure below). We found that, as expected from our previous analysis, the number of clusters was not correlated to nucleus size. S2 cells displayed a large heterogeneity of nuclear volumes and number of clusters, whereas the distributions were narrower for embryonic cells. A linear regression between number of clusters and nuclear volumes could only explain 21% of the variation for S2 cells ($r^2 = 0.209$), 6% for late embryo ($r^2 = 0.061$), and fails completely for early embryo ($r^2 = -0.083$).

Interestingly, for the smallest nuclei detected for each cell type, the number of spots was often above the mean: for instance, in early embryo many cells displayed more than 40 clusters (similar observations can be made for late embryo and S2 cells). This indicates that the spatial resolution of 3D-SIM is enough to resolve clusters even in the smallest cells. On the other hand, the largest nuclei often displayed very low number of spots when compared to the mean of each particular cell type, confirming the large degree of stochasticity in chromatin organization independently of volume size.

This new analysis is now included in the extended data of the revised version of the manuscript (Supplementary Fig. 3 panel f) along with a discussion of our interpretation.

The revised version of the main text (pages 9-10) now reads:

'The lower number of foci detected was not associated with the smaller volume of embryonic cell nuclei causing the probes to be closer than the resolution limit of 3D-SIM microscopy (Supplementary Fig. 3e). Furthermore, for each cell type, the number of foci displayed very low or nonexistent correlation with nucleus size (Supplementary Fig. 3f).'

Legend of Supplementary Fig. 3f: Number of detected spots as a function of nuclear volume for each cell. Mean volume is $(32\pm 13) \mu\text{m}^3$, (35 ± 17) and $(67\pm 23) \mu\text{m}^3$ for late embryo, early embryo and S2 cells, respectively. $N=180$.

2.) Does an exponential model, or a power-law model, best fit the data comparing Hi-C contact frequency to absolute contact probability?

Answer:

We have fitted the dataset with a power-law and an exponential model. We found that both empirical relations fit the data equally well (see Figure below). We have now modified Fig. 2a to show both fits and modified the text accordingly.

The revised text now reads (page 7):

"Hi-C contact frequencies among TAD borders increased nonlinearly with absolute contact probabilities (Fig. 2a and Supplementary Fig. 2a), with both exponential and power-law empirical models fitting the data equally well."

Legend of Fig. 2a: Normalized Hi-C counts vs. microscopy absolute contact probability for consecutive and non-consecutive domain borders for embryo and S2 cells. Solid black and red lines represent an exponential and a power-law fits respectively.

3.) While the differences between S2 and embryonic cells are striking, it is worth noting that these differences could be between cell types or between primary and immortalized cells. In fact, late stage and early stage embryonic samples appear more similar than they are different; without another cell line or another primary tissue it is difficult to say that this is truly a developmental difference and not amplified by the process of immortalization and cell culture.

Answer:

We agree with the referee. In the revised version of the manuscript, we are more cautious and explicitly mention that the differences observed between S2 and embryonic cells likely reflect differences in cell-type-specific transcriptional programs. We do not attribute any longer these differences as arising from developmental differences.

The new version of the text now reads (page 13):

'... these results are consistent with the cell-type specific higher-order organization of chromatin arising from stochastic contacts between chromosomal regions harboring similar epigenetic marks, likely reflecting cell-type specific transcriptional programs.'

4.) What is the coefficient of variation for distance between two regions? How does it vary between cell types? Since standard deviation scales linearly with mean distance

(Supplementary Fig 1i), it is likely to be constant, and would be a useful quantifier for the observed stochasticity.

Answer:

To answer this remark, we have calculated the coefficient of variation (CV) as a function of physical distance for the different cell types (see below and new Supplementary Fig. 1j). As predicted by the reviewer, we observe that CV is constant, and independent of physical distance between probes for all cell types. Consistently with Supplementary Fig. 1i, we observe that the CV is larger for late embryos than for S2 cells.

New Supplementary Fig. 1j: Coefficient of variation for physical distances between borders as a function of the mean physical distance for all three cell types. Dotted lines represent the average for late embryo (0.74, light blue) and S2 cells (0.65, orange).

Overall, we think this is a well-done and interesting study with highly suggestive implications for the field.

We thank the reviewer for her/his comments and suggestions that have significantly improved the clarity and solidity of our work.

Reviewer #3:

General comment

In this manuscript, Cattoni and colleagues attempt to compare physical distances between certain regions of the genome in the nucleus using super-resolution imaging, coupled to fluorescence in situ hybridization and immunofluorescence, with high-resolution Hi-C data, in Drosophila embryos at two developmental stages (early/stage 5 and late/stage 16) and in cultured S2 cells.

They labelled TAD boundaries using FISH and imaged cells using 3D-SIM to assess the variability in TAD organization between single cells. While this is a valid and a interesting approach, there are some points that need to be better defined, discussed or analysed. In particular, the discussion of models of TAD formation is incomplete. First of all, the most popular and current loop extrusion model for TAD formation (at least in mammalian cells) is not discussed. This is important because this model does not require stable looping of TAD boundaries – a central assumption of this manuscript – instead TADs arise from a dynamic balance between cohesion-mediated loop extrusion, the blocking of that movement by architectural proteins and activities that either promote loop extrusion by cohesion (Nipbl) or that remove cohesin from the chromosome (WAPL). Are the authors data consistent with this model? A model of TAD boundaries in the fly genome, based on simple nucleosome properties - especially at active genes (Ulianov et al, Genome Research, 2016) is also not discussed.

Answer:

We apologize for the lack of discussion of models of TAD formation. We have now amended our discussion section to address this concern of the reviewer. The relevant section in pages 13-14 now reads:

“In mammals, a large proportion of consecutive TAD borders are looped by specific interactions apparently mediated by CTCF and cohesin (Rao et al. 2015; Sanborn et al. 2015; Vietri Rudan et al. 2015). Recent reports suggested that this mechanism may also be at play in Drosophila (Hug et al. 2017; Li et al. 2015). Our results, however, provide compelling evidence that looping of consecutive TADs borders in Drosophila is rare at the single-cell level. These observations, supported by recent studies showing that cohesin-enriched loop anchors in Drosophila are found within TADs rather than at TAD borders (Cubeñas-Potts et al. 2016; Eagen et al. 2017), are against TAD boundaries being the bases of stable chromatin loops. Thus, the lack of frequent interactions between TAD borders could be consistent with a model where TADs arise from a dynamic balance between cohesin-mediated loop extrusion (Fudenberg et al. 2016), the blocking of that movement by architectural proteins, and factors that may load or remove cohesin (Haarhuis et al. 2017; Barrington et al. 2017). In Drosophila, however, CTCF and cohesin are not found enriched at TAD borders. Thus, we envision that other factors (e.g. Beaf-32/GAF and CP190/chromator instead of CTCF and cohesin) could play a role at looping and dynamically extruding distant DNA fragments within the same TAD. In addition, active marks may help determine the properties of TAD boundaries (Ulianov et al. 2016) while other epigenetic marks could play a role in the formation of polycomb and inactive

TADs (Jost et al. 2014). Similar epigenetic mechanisms may even play a role in TAD folding in mammals, consistent with the observation that CTCF depletion leads only to minor changes in TAD organization (Zuin et al. 2014). Importantly, our data provides quantitative estimates of the stochasticity and absolute frequencies of interactions within TADs, imposing important constraints on any model of TAD formation in *Drosophila*.”

The authors place a lot of emphasis on the stochastic nature of chromatin organization, without a proper citation of the well-studied levels of stochastic genome organization in the literature (e.g. association with the nuclear lamina, chromosome territory neighborhoods, gene positioning within chromosome territories).

Answer:

We apologize for not having cited these important references. These are now cited in the Introduction section. We will be happy to cite other studies should the reviewer find it necessary. The relevant section of the introduction now reads (page 3):

“These structural levels are often seen as highly stable over time, however, recent single-cell Hi-C studies have reported different degrees of heterogeneity (Giorgetti et al. 2014; Flyamer et al. 2017). Other studies have reported that genomes also display stochasticity in their association with the nuclear lamina (Kind & van Steensel 2014), in the formation of chromosome territory neighborhoods (Cremer & Cremer 2010), and in gene kissing (Lanctôt et al. 2007)”

Specific comments

1.) Are the probes located inside the TAD (adjacent to its boundary), overlapping the TAD boundary, or just outside (in the inter-TAD or the adjacent TAD)?

Answer:

The probes overlap the boundary between TADs. All probes were designed taking as reference the center of the barriers and with the minimal size that ensured optimal imaging conditions without extending into the interior of TADs.

2.) In Fig. 1C the boxed legend is labeled incorrectly. Presumably filled circles should be TB2-TB3, not TB2-TB2

Answer:

The label in the box legend has been corrected.

3.) In Fig. 2D, it is more common to show this kind of plot in log-log scale (with log spaced bins on x axis), this way power-law dependencies look like straight lines and their slope can be more

easily estimated.

Answer:

We followed the advice of the referee and now plot Fig. 2D in log-log scale. The new panel is presented below:

4.) *In Fig. 2b and S2b it is unclear what was taken as the physical distance – was it the mean of the distance between a pair of probes? Then, perhaps, error bars for this measurement need to be shown too?*

Answer:

The reviewer's observation is correct, in Figs 2b and Supplementary Fig. 2b the mean physical distance between probes is displayed in the horizontal axis. To improve the clarity of the legend we have now employed the expression 'mean physical distance between probes'. We agree with the reviewer that error bars should be displayed in the figures. In Supplementary Fig. 2b, we have now included the error bars for the physical distance. However, in the main figure we have decided to remove the error bars, otherwise the visibility of the data is very compromised. As the set of data of Figs. 2b and Supplementary Fig. 2b are the same, the error bars can always be found by an interested reader in Supplementary Fig. 2b.

In the revised version, the legend of Fig. 2b now reads:

'Absolute contact probability vs. mean physical distance between probes for consecutive and non-consecutive TAD borders (filled circles). Solid lines represent power-law fittings with scaling exponents described in Supplementary Fig. 2b. Triangles represent measurements within TADs.'

5.) *It would be interesting to directly compare (and discuss differences, if there are any) power-law scaling exponents between Hi-C data and FISH data, since analysis of contact*

probability decay is a common way to assay polymer properties of the genome from Hi-C data. And since the authors have access to physical distance measurements between pairs of loci at many separations, a discussion of chromatin polymer properties based on this would be warranted – i. e. whether the resultant curve is consistent with existing models of chromatin organization.

Answer:

We have reflected several times before the initial submission on whether FISH power-law exponents can be directly compared to power-law exponents derived from Hi-C data, and on whether FISH exponents can be used to assay polymer properties of the model. We have since embarked on exchanges with experts on polymer models of the genome, which confirmed that these comparisons are not straightforward. Thus, we would prefer to be cautious in our interpretation of these power-law coefficients and straight comparisons. Future studies will be required to integrate our data with Hi-C datasets using a comprehensive theoretical framework to further understand the polymer properties of the genome.

In the results section (page 8), we added a short, cautious discussion of power-law exponents derived from our experiments. This discussion is repeated below:

“Theoretical studies of polymer physics suggest that the exponent of polymers with random coil behaviour is $\frac{1}{2}$, while that of an equilibrium globule is $\frac{1}{3}$ (Mirny 2011). Thus, our power-law exponents situate between these two extremes, suggesting an intermediate behaviour.”

6. In Fig. 3b, after labelling 69 foci on chromosome 3R the authors obtained an average of 89 spots in S2 cells, a lot more than expected – is it perhaps due to them having 4 copies of each chromosome, or chromosome rearrangements in the population? The latter could also explain a somewhat bimodal distribution of distances in Fig. 3B if the rearrangements are only present in a part of the population, but both explanations unfortunately would invalidate the whole comparison of this cell line to embryos with this approach.

Answer:

As the Reviewer pointed out, embryonic cells are diploid while S2 cells are tetraploid. We observed a similar number of spots per cell when imaging single borders for all cell types (Supplementary Fig. 1c), indicating a similarly high degree of homologous chromosome pairing for all cell types in spite of different ploidy. To estimate the number of expected foci after labeling 69 barriers in different cell types, we took into account the proportions of cells with 1, 2 and 3 foci (~65% display a single focus, ~30% two foci, and ~5% three foci, Supplementary Fig. 1c). In this manner, we can estimate that 69, non-interacting barriers should display in average 97 foci/cell in S2 cells, consistent with our experimental finding of 89 ± 28 foci/cell, and validating our approach. With a similar calculation we expect in average 92 and 100 foci/cell for late and early embryo, respectively. Surprisingly, we observed a much lower number of spots per cell than expected in these two cell types (51 ± 20 for early and 36 ± 13 for late embryos).

These results are then consistent with embryonic cells displaying long-range interactions between borders.

This rationale was previously discussed in Supplementary Fig. 3, but now we have also included a brief description in the main text (page 9):

“From the number of labeled barriers (69) and the pairing frequency of homologous chromosomes (Supplementary Fig. 1c), we can estimate a maximum of 90-100 resolvable foci/cell in the absence of any long-range interactions (Supplementary Fig. 3). Our imaging results show an average of 89 ± 28 foci/cell for S2 cells (Fig. 3b), confirming our predictions and consistent with a very low frequency of long-range interactions for this cell type (see discussion in Supplementary Fig. 3). Surprisingly, in early and late embryos the number of observed foci was considerably reduced (51 ± 20 for early and 36 ± 13 , respectively, and Fig. 3b), revealing higher probabilities of long-range interactions for these cell-types.”

7. In the STORM analysis of compartment clustering, how does the proportion of clusters unaccounted for from modelling based on 1D distribution of histone marks correspond to frequency of interactions between the corresponding regions by Hi-C? One would expect to see higher interaction frequencies between K27me3 regions in the data from embryos than from S2 cells (and the same for K4me3), and this would be a good validation of the approach. And a similar question can be asked about compartment densities (Fig. S6a) – do FISH measurements correlate with Hi-C data, e.g. when comparing number of contacts inside K27me3 and K4me3 domains?

Answer:

To address this remark, we performed two new analyses. First, we calculated the distribution of Hi-C contact frequency between H3K27me3 domains in embryos and S2 cells. We observed that embryos displayed higher interaction frequencies than S2 cells, as expected by the reviewer, and consistent with our STORM data. Similar results were obtained for H3K4me3. These data are now presented in a new panel of Figure 4 (Fig. 4i) and shown below.

The revised text now reads (page 12):

“...The latter is consistent with higher Hi-C contact frequency between K27me3 domains in embryos than in S2 cells (Fig. 4i).”

Fig. 4i. Box plots of the distributions of normalised Hi-C counts between chromatin domains of the same type (H3K27me3 or H3K4me3) in embryos and S2 cells. These results were independent of matrix resolution (10, 20 and 50 kb). Boxes contain 50% of the data (0.67σ), and red lines mark the median values. Outliers ($>3.3\sigma$ away from the mean values) are shown as black dots. P-values were calculated using the Welch t-test.

Secondly, we measured the mean number of Hi-C contacts inside H3K27me3 and H3K4me3 domains. This analysis shows that H3K27me3 domains display in average more contacts than K4me3 domains in both embryos and S2 cells. This result is consistent with our STORM data showing that compartment densities are higher for H3K27me3 than for H3K4me3 in all cell types investigated. These new analysis are now presented in a new panel in Supplementary Figure 6b and shown below.

The text have been changed accordingly (page 11):

‘Notably, the local density of compartments was higher for both types of marks in embryonic cells than for S2 cells (Supplementary Fig. 6a), consistent with our previous findings (Figs. 2e-f) and with compartment contact density from Hi-C counts (Supplementary Fig. 6b).’

Supplementary Fig. 6b. Boxplot of the distribution of relative Hi-C normalised counts (observed/expected) within H3K27me3 or H3K4me3 domains in embryos and S2 cells. Each entry of the Hi-C normalised matrix has been divided by the genome-wide average normalised Hi-C counts at the corresponding genomic distance to take into account for the expected diagonal decay of the Hi-C data. We found the results to be robust over various matrix resolutions (10, 20 and 50 kb). Boxes contain 50% of the data (0.67σ), and the red lines inside them mark the medians. Outliers ($>3.3\sigma$ away from the mean values) are shown as black dots. P-values were calculated using the Welch t-test.

References

- Barrington, C., Finn, R. & Hadjur, S., 2017. Cohesin biology meets the loop extrusion model. *Chromosome research: an international journal on the molecular, supramolecular and evolutionary aspects of chromosome biology*, 25(1), pp.51–60.
- Boettiger, A.N. et al., 2016. Super-resolution imaging reveals distinct chromatin folding for different epigenetic states. *Nature*. Available at: <http://dx.doi.org/10.1038/nature16496>.
- Cattoni, D.I. et al., 2013. Super-resolution imaging of bacteria in a microfluidics device. *PLoS one*, 8(10), p.e76268.
- Contrino, S. et al., 2012. modMine: flexible access to modENCODE data. *Nucleic acids research*, 40(Database issue), pp.D1082–8.
- Cremer, T. & Cremer, M., 2010. Chromosome territories. *Cold Spring Harbor perspectives in biology*, 2(3), p.a003889.
- Cubeñas-Potts, C. et al., 2016. Different enhancer classes in *Drosophila* bind distinct architectural proteins and mediate unique chromatin interactions and 3D architecture. *Nucleic acids research*. Available at: <http://dx.doi.org/10.1093/nar/gkw1114>.
- Culley, S. et al., 2017. NanoJ-SQUIRREL: quantitative mapping and minimisation of super-resolution optical imaging artefacts. *bioRxiv*. Available at: <http://www.biorxiv.org/content/early/2017/07/02/158279.abstract>.
- Eagen, K.P., Aiden, E.L. & Kornberg, R.D., 2017. Polycomb-mediated chromatin loops revealed by a subkilobase-resolution chromatin interaction map. *Proceedings of the National Academy of Sciences of the United States of America*. Available at: <http://dx.doi.org/10.1073/pnas.1701291114>.
- Faure, L.M. et al., 2016. The mechanism of force transmission at bacterial focal adhesion complexes. *Nature*. Available at: <http://dx.doi.org/10.1038/nature20121>.
- Fiche, J.-B. et al., 2013. Recruitment, assembly, and molecular architecture of the SpoIIIE DNA pump revealed by superresolution microscopy. *PLoS biology*, 11(5), p.e1001557.
- Flyamer, I.M. et al., 2017. Single-nucleus Hi-C reveals unique chromatin reorganization at oocyte-to-zygote transition. *Nature Publishing Group*. Available at: <http://dx.doi.org/10.1038/nature21711>.
- Fudenberg, G. et al., 2016. Formation of Chromosomal Domains by Loop Extrusion. *Cell reports*, 15(9), pp.2038–2049.
- Georgieva, M. et al., 2014. Nanometer resolved single-molecule colocalization of nuclear factors by two-color super resolution microscopy imaging. *Methods*. Available at: <http://www.sciencedirect.com/science/article/pii/S1046202316300639>.
- Georgieva, M. et al., 2016. Nanometer resolved single-molecule colocalization of nuclear factors by two-color super resolution microscopy imaging. *Methods*, 105, pp.44–55.

- Giorgetti, L. et al., 2014. Predictive polymer modeling reveals coupled fluctuations in chromosome conformation and transcription. *Cell*, 157(4), pp.950–963.
- Haarhuis, J.H.I. et al., 2017. The Cohesin Release Factor WAPL Restricts Chromatin Loop Extension. *Cell*, 169(4), pp.693–707.e14.
- Hug, C.B. et al., 2017. Chromatin Architecture Emerges during Zygotic Genome Activation Independent of Transcription. *Cell*, 169(2), pp.216–228.e19.
- Jost, D. et al., 2014. Modeling epigenome folding: formation and dynamics of topologically associated chromatin domains. *Nucleic acids research*, 42(15), pp.9553–9561.
- Kind, J. & van Steensel, B., 2014. Stochastic genome-nuclear lamina interactions: modulating roles of Lamin A and BAF. *Nucleus*, 5(2), pp.124–130.
- Lanctôt, C. et al., 2007. Dynamic genome architecture in the nuclear space: regulation of gene expression in three dimensions. *Nature reviews. Genetics*, 8(2), pp.104–115.
- Le Gall, A. et al., 2016. Bacterial partition complexes segregate within the volume of the nucleoid. *Nature Communications*, in press ().
- Li, L. et al., 2015. Widespread rearrangement of 3D chromatin organization underlies polycomb-mediated stress-induced silencing. *Molecular cell*, 58(2), pp.216–231.
- Marbouty, M. et al., 2015. Condensin- and Replication-Mediated Bacterial Chromosome Folding and Origin Condensation Revealed by Hi-C and Super-resolution Imaging. *Molecular cell*, 59(4), pp.588–602.
- Mirny, L.A., 2011. The fractal globule as a model of chromatin architecture in the cell. *Chromosome research: an international journal on the molecular, supramolecular and evolutionary aspects of chromosome biology*, 19(1), pp.37–51.
- Rao, S.S.P. et al., 2015. A 3D Map of the Human Genome at Kilobase Resolution Reveals Principles of Chromatin Looping. *Cell*, 162(3), pp.687–688.
- Ricci, M.A. et al., 2015. Chromatin fibers are formed by heterogeneous groups of nucleosomes in vivo. *Cell*, 160(6), pp.1145–1158.
- Sanborn, A.L. et al., 2015. Chromatin extrusion explains key features of loop and domain formation in wild-type and engineered genomes. *Proceedings of the National Academy of Sciences of the United States of America*, 112(47), pp.E6456–65.
- Schneider, I., 1972. Cell lines derived from late embryonic stages of *Drosophila melanogaster*. *Journal of embryology and experimental morphology*, 27(2), pp.353–365.
- Sergé, A. et al., 2008. Dynamic multiple-target tracing to probe spatiotemporal cartography of cell membranes. *Nature methods*, 5(8), pp.687–694.
- Sexton, T. et al., 2012. Three-dimensional folding and functional organization principles of the *Drosophila* genome. *Cell*, 148(3), pp.458–472.

- Tarancón Díez, L. et al., 2014. Coordinate-based co-localization-mediated analysis of arrestin clustering upon stimulation of the C-C chemokine receptor 5 with RANTES/CCL5 analogues. *Histochemistry and cell biology*, 142(1), pp.69–77.
- Tokunaga, M., Imamoto, N. & Sakata-Sogawa, K., 2008. Highly inclined thin illumination enables clear single-molecule imaging in cells. *Nature methods*, 5(2), pp.159–161.
- Ulianov, S.V. et al., 2016. Active chromatin and transcription play a key role in chromosome partitioning into topologically associating domains. *Genome research*, 26(1), pp.70–84.
- Van Bortle, K. & Corces, V.G., 2013. The role of chromatin insulators in nuclear architecture and genome function. *Current opinion in genetics & development*, 23(2), pp.212–218.
- Vietri Rudan, M. et al., 2015. Comparative Hi-C reveals that CTCF underlies evolution of chromosomal domain architecture. *Cell reports*, 10(8), pp.1297–1309.
- Zuin, J. et al., 2014. Cohesin and CTCF differentially affect chromatin architecture and gene expression in human cells. *Proceedings of the National Academy of Sciences of the United States of America*, 111(3), pp.996–1001.

REVIEWERS' COMMENTS:

Reviewer #1 (Remarks to the Author):

The authors have addressed all my remaining concerns. Thus I am happy to recommend publication of this manuscript in Nature Communications.

Reviewer #2 (Remarks to the Author):

The authors have satisfactorily addressed all our points. We recommend publication.

Reviewer #3 (Remarks to the Author):

The authors have done a very thorough job in responding to my comments and those of the other reviewers. I would be happy to see this paper published in Nature Communications.